# Variational Counterfactual Intervention Planning to Achieve Target Outcomes

Xin Wang [1]  Shengfei Lyu [2]  Chi Luo [1]  Xiren Zhou [1]  Huanhuan Chen [1]

## Abstract

A key challenge in personalized healthcare is identifying optimal intervention sequences to guide temporal systems toward target outcomes, a novel problem we formalize as counterfactual target achievement. In addressing this problem, directly adopting counterfactual estimation methods face compounding errors due to the unobservability of counterfactuals. To overcome this, we propose Variational Counterfactual Intervention Planning (VCIP), which reformulates the problem by modeling the conditional likelihood of achieving target outcomes, implemented through variational inference. By leveraging the g-formula to bridge the gap between interventional and observational log-likelihoods, VCIP enables reliable training from observational data. Experiments on both synthetic and real-world datasets show that VCIP significantly outperforms existing methods in target achievement accuracy.

## 1. Introduction

Causal inference in time series data plays a crucial role in healthcare and clinical decision support (Wu et al., 2022; Zheng et al., 2020; Prosperi et al., 2020). With the widespread adoption of electronic health records (EHRs), practitioners face the challenge of evaluating long-term treatment effects on patient outcomes, making accurate causal analysis essential for informed clinical decisions (Lim et al., 2018; Jensen et al., 2012; Chakraborty et al., 2022).

Existing research in temporal causal inference primarily focuses on the "forward prediction" problem of predicting future system trajectories given historical observations and intervention sequences (Lim et al., 2018; Bica et al., 2020; Melnychuk et al., 2022; Wang et al., 2024). While this

paradigm has advanced understanding of intervention effects, decision-makers often face a critical inverse problem of identifying optimal intervention sequences that guide systems toward desired target outcomes.

This paper introduces the counterfactual target achievement problem, a novel sequential decision-making problem that aims to find intervention sequences guiding system outputs toward specific target outcomes based on historical observations. Although similar to the sequential decision-making framework in dynamic treatment regimes (DTRs) (Laber et al., 2014), this problem differs fundamentally in its objective. While DTRs aim to maximize population-level outcomes, decision-makers in many contexts may focus on achieving specific target outcomes for individual patients. For example, in diabetes management, physicians adjust insulin interventions to maintain each patient's blood glucose within a personalized target range, rather than optimizing glycemic control for the population (Bergenstal, 2018; Peterson et al., 2007).

An intuitive solution is to leverage counterfactual estimation methods (e.g., CRN (Bica et al., 2020)) to estimate potential outcomes under different intervention sequences and select interventions by comparing these outcomes to the target outcome. However, as shown in the left panel of Figure 1, while sequence 2 is predicted to be closer to the target (blue dashed line), the true outcome (blue solid line) deviates significantly. This illustrates a key challenge: since counterfactuals are unobservable in practice, prediction errors cannot be accurately assessed, leading to biases in estimated target distances and misleading intervention selections. Indeed, sequence 1 achieves better true performance (red solid line) despite its predicted trajectory (red dashed line) appearing less favorable.

To address these limitations, we reformulate the counterfactual target achievement problem by directly modeling the probability of reaching the target outcome, conditioned on the observed history, as shown in the right panel of Figure 1. By introducing latent variables to capture temporal system state evolution, we propose the Variational Counterfactual Intervention Planning (VCIP) framework, which avoids the pitfalls of counterfactual estimation methods. VCIP first constructs an evidence lower bound (ELBO), a tool used for optimizing likelihoods under interventional distributions,

[1]University of Science and Technology of China
[2]Nanyang Technological University. Correspondence to: Huanhuan Chen <hchen@ustc.edu.cn>, Shengfei Lyu <shengfei.lyu@ntu.edu.sg>.

*Proceedings of the $42^{nd}$ International Conference on Machine Learning*, Vancouver, Canada. PMLR 267, 2025. Copyright 2025 by the author(s).

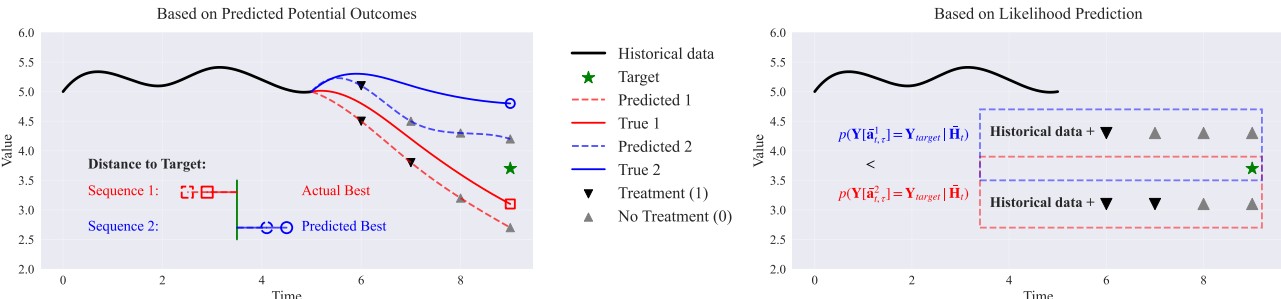

*Figure 1.* Illustration of two prediction methods for counterfactual target achievement. Left: The counterfactual estimation method directly estimates potential outcomes under different intervention sequences, showing both predicted (dashed lines) and true (solid lines) trajectories, with distance indicators demonstrating each sequence's closeness to the target outcome. Right: VCIP evaluates intervention sequences by modeling the probability of achieving the target outcome, conditioned on the observed history, and optimizing the likelihood of reaching the target.

and leverages the g-formula (Robins, 1986) to establish connections between interventional and observational likelihoods. VCIP circumvents error accumulation that typically arises from relying on explicit predictions and offers a robust training framework, ensuring reliability with observational data. Our main contributions are summarized as follows:

- We formulate the novel temporal counterfactual target achievement problem of identifying intervention sequences that guide dynamic systems toward specified target outcomes.

- We propose Variational Counterfactual Intervention Planning (VCIP), a framework that directly models target achievement likelihood through variational inference, with theoretical guarantees for training on observational data.

- Through extensive experiments on both synthetic and real-world datasets, we demonstrate that VCIP consistently outperforms existing approaches in terms of target achievement accuracy.

## 2. Related Work

### 2.1. Temporal Counterfactual Estimation

Estimating temporal counterfactual outcomes is closely related to but fundamentally different from the proposed target achievement task. Early work in this field focused on statistical methods including g-formula, structural nested models, and marginal structural models (Robins, 1986; Robins et al., 2000; Robins & Hernán, 2009). Due to the limitations of linear regression in handling complex temporal dependencies (Mortimer et al., 2005), research gradually shifted towards Bayesian nonparametric frameworks (Xu et al., 2016; Soleimani et al., 2017; Roy et al., 2017). How-

ever, these methods imposed strong prior assumptions on model structures, limiting their practical applicability.

Recent advances primarily leverage deep neural networks, including RMSN (Lim et al., 2018), CRN (Bica et al., 2020), G-Net (Li et al., 2021), CT (Melnychuk et al., 2022), and ACTIN (Wang et al., 2024). Architecturally, while RMSN, CRN, and G-Net build upon basic LSTM networks, CT adopts the more powerful Transformer architecture (Vaswani et al., 2017) to address limitations in capturing long-range dependencies among time-varying confounders in longitudinal data (Hochreiter et al., 2001). ACTIN further proposes a dual-module architecture to balance effectiveness and efficiency.

These methods employ time-varying adjustment techniques to mitigate confounding bias. RMSN builds dual propensity networks with IPTW scoring. G-Net integrates g-formula methods. CRN, CT, and ACTIN learn balanced representations through gradient reversal, reverse KL divergence minimization, and distribution-based adversarial training respectively. While these approaches advance temporal counterfactual estimation, they have limitations for the target achievement task.

### 2.2. Dynamic Treatment Regimes

Dynamic treatment regimes (DTRs) provide a formal framework for personalizing treatments in sequential medical decision-making while optimizing the population-level outcomes (Murphy, 2003). Most existing work in DTR optimization focuses on offline learning from observational data (Zhang & Bareinboim, 2019). Recent works have explored reinforcement learning (RL) approaches for DTR optimization (Luckett et al., 2020; Raghu et al., 2017; Zhang & Bareinboim, 2019), addressing challenges such as policy evaluation biases from confounded data (Luo et al., 2024)

and continuous state spaces (Tsirtsis & Gomez-Rodriguez, 2023). However, these methods primarily focus on finding optimal treatment policies at the population level while maintaining individual-level adaptivity. In contrast, our work aims to predict and optimize treatment sequences for specific individuals based on their historical trajectories, differing from DTRs in two key aspects: (1) we focus on individual-level optimization rather than population-level policy learning, and (2) we directly maximize the probability of achieving patient-specific target outcomes instead of optimizing expected outcomes across the population.

## 3. Problem Formulation

Let $\mathcal{D}$ represent a longitudinal observational dataset comprising records from $N$ subjects, structured as $\mathcal{D} = \left\{ \{\mathbf{x}_t^{(i)}, \mathbf{a}_t^{(i)}, \mathbf{y}_t^{(i)}\}_{t=1}^{T^{(i)}} \cup \{\mathbf{v}^{(i)}\} \right\}_{i=1}^{N}$. Each subject's trajectory, denoted by index $i$, consists of sequential measurements over $T^{(i)}$ time points, where $\mathbf{X}_t^{(i)} \in \mathcal{X}$ represents time-varying covariates, $\mathbf{A}_t^{(i)} \in [0,1]^d$ or $\{0,1\}^d$ denotes continuous-valued or binary treatments, respectively, and $\mathbf{Y}_t^{(i)} \in \mathcal{Y}$ indicates the measured outcomes. Time-invariant characteristics are captured in $\mathbf{V}^{(i)} \in \mathcal{V}$. For simplicity in notation, we will drop the subject indicator $(i)$ when the context is clear.

The temporal progression of a subject can be captured through their historical trajectory $\bar{\mathbf{H}}_t = (\bar{\mathbf{X}}_t, \bar{\mathbf{A}}_{t-1}, \bar{\mathbf{Y}}_t, \mathbf{V})$, where the bar notation represents the sequence up to time $t$: $\bar{\mathbf{X}}_t = (\mathbf{X}_1, \cdots, \mathbf{X}_t)$, $\bar{\mathbf{Y}}_t = (\mathbf{Y}_1, \cdots, \mathbf{Y}_t)$, and $\bar{\mathbf{A}}_{t-1} = (\mathbf{A}_1, \cdots, \mathbf{A}_{t-1})$.

Under the potential outcomes framework (Rubin, 1978), let $\bar{\mathbf{a}}_{t,\tau} = (\mathbf{a}_t, \cdots, \mathbf{a}_{t+\tau-1})$ denote a sequence of treatments from time $t$ to $t + \tau - 1$. We denote $\mathbf{Y}[\bar{\mathbf{a}}_{t,\tau}]$ as the potential outcome at time $t + \tau$ that would be observed under treatment sequence $\bar{\mathbf{a}}_{t,\tau}$ (Robins et al., 2000). For counterfactual target achievement, given a desired future state $\mathbf{Y}_{\text{target}}$ at time $t + \tau$, our research question focuses on determining the optimal sequence of treatments that maximizes the likelihood of achieving this target:

$$\bar{\mathbf{a}}^* = \operatorname{argmax}_{\bar{\mathbf{a}}} \, p(\mathbf{Y}[\bar{\mathbf{a}}_{t,\tau}] = \mathbf{Y}_{\text{target}} \mid \bar{\mathbf{H}}_t). \quad (1)$$

For causal identification with observational data, we rely on standard assumptions including consistency, sequential ignorability, and positivity (see Appendix A). For comparison of counterfactual target achievement with counterfactual estimation and DTRs, we refer readers to Appendix C.

## 4. Methodology

In this work, we adopt a causal model as illustrated in Figure 2. The model incorporates latent factors $\mathbf{Z}_s$ at each timestep $s$ ($s = t, \cdots, t + \tau$) to represent the underlying

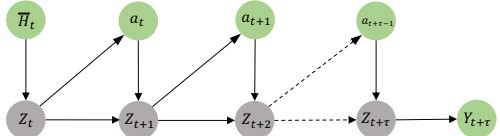

*Figure 2.* Causal model for temporal dynamics. The latent factor $\mathbf{Z}_s$ at time $s$ represents the system state, which encapsulates the historical information $\bar{\mathbf{H}}_s$. Specifically, $\mathbf{Z}_t$ is encoded from historical information $\bar{\mathbf{H}}_t$, and the outcome $\mathbf{Y}_{t+\tau}$ is decoded from the final latent state $\mathbf{Z}_{t+\tau}$. The dashed arrows indicate the omitted intermediate steps in the temporal sequence.

system state. We assume that $\mathbf{Z}_s$ encodes all relevant historical information contained in $\bar{\mathbf{H}}_s$, thus serving as a sufficient representation of the system's temporal evolution.

### 4.1. Evidence Lower Bound (ELBO) Derivation

The optimal intervention sequence for Equation 1 is obtained by maximizing:

$$\mathcal{O} := \log p_\theta(\mathbf{Y}[\bar{\mathbf{a}}_{t,\tau}] = \mathbf{Y}_{t+\tau} \mid \bar{\mathbf{H}}_t), \quad (2)$$

where $\theta$ denotes the parameters of the generative model. Following the causal model in Figure 2, we introduce a variational distribution $q_\phi(\bar{\mathbf{Z}}_{t,\tau+1} \mid \bar{\mathbf{H}}_t, \mathbf{Y}_{t+\tau}, \bar{\mathbf{a}}_{t,\tau})$ to derive the ELBO for optimizing $\mathcal{O}$:

$$\text{ELBO}_1 = \mathbb{E}_{q_\phi}[\log \frac{p_\theta(\mathbf{Y}_{t+\tau}, \bar{\mathbf{Z}}_{t,\tau+1} \mid \bar{\mathbf{H}}_t, do(\bar{\mathbf{a}}_{t,\tau}))}{q_\phi(\bar{\mathbf{Z}}_{t,\tau+1} \mid \bar{\mathbf{H}}_t, \mathbf{Y}_{t+\tau}, \bar{\mathbf{a}}_{t,\tau})}], \quad (3)$$

where the $do(\cdot)$ operator (Pearl, 2009) represents atomic interventions that set variables to specific values, breaking the causal link from $\mathbf{Z}_s$ to $\mathbf{a}_s$ in the original causal model. As a practical approximation, due to the inaccessibility of the intervention distribution, we substitute it with the observational distribution. Therefore, we introduce a variational distribution $q_\phi(\bar{\mathbf{Z}}_{t,\tau+1} \mid \bar{\mathbf{H}}_t, \mathbf{Y}_{t+\tau}, \bar{\mathbf{a}}_{t,\tau})$ to approximate the posterior distribution under intervention, where the effects of interventions are partially captured in the observed outcomes $\mathbf{Y}_{t+\tau}$.

However, in our causal model, the intervention assignment depends on latent factors, making it infeasible to directly obtain the interventional distribution and optimize $\text{ELBO}_1$. Instead, we can optimize the following log-likelihood objective based on observational data:

$$\mathcal{O}' := \log p_\theta(\mathbf{Y}_{t+\tau} \mid \bar{\mathbf{H}}_t, \bar{\mathbf{a}}_{t,\tau}) \quad (4)$$

Similarly, the ELBO for optimizing $\mathcal{O}'$ can be derived as:

$$\text{ELBO}_2 = \mathbb{E}_{q_\phi}[\log \frac{p_\theta(\mathbf{Y}_{t+\tau}, \bar{\mathbf{Z}}_{t,\tau+1} \mid \bar{\mathbf{H}}_t, \bar{\mathbf{a}}_{t,\tau})}{q_\phi(\bar{\mathbf{Z}}_{t,\tau+1} \mid \bar{\mathbf{H}}_t, \mathbf{Y}_{t+\tau}, \bar{\mathbf{a}}_{t,\tau})}]. \quad (5)$$

To bridge the gap between interventional and observational log-likelihoods ($\mathcal{O}$ and $\mathcal{O}'$), we establish the following theorem:

**Theorem 4.1.** *Given the causal model illustrated in Figure 2, assume there exists constants $\epsilon_1, \epsilon_2 > 0$ such that:*

$$\mathcal{O} - \mathrm{ELBO}_1 \leq \epsilon_1, \quad \mathcal{O}' - \mathrm{ELBO}_2 \leq \epsilon_2.$$

*Then optimizing $\mathcal{O}$ can be approximated by optimizing:*

$$\mathrm{ELBO}_2 - \sum_{s=t}^{t+\tau-1} \mathbb{E}_{q_\phi}[\log p_\theta(\mathbf{a}_s \mid \mathbf{Z}_s)] + \log p_\theta(\bar{\mathbf{a}}_{t,\tau} \mid \bar{\mathbf{H}}_t) \quad (6)$$

*with the error bounded by $\epsilon_1 + \epsilon_2$.*

The key to proving Theorem 4.1 lies in leveraging the G-formula (Hedeker & Gibbons, 2006), which states:
*Property* 1 (g-formula). The expected potential outcome under intervention sequence $\bar{\mathbf{a}}_{t,\tau}$ can be expressed as:

$$p_\theta(\mathbf{Y}[\bar{\mathbf{a}}_{t,\tau}] = \mathbf{Y}_{t+\tau} | \bar{\mathbf{H}}_t) =$$
$$\int p_\theta(\mathbf{Y}_{t+\tau} | \bar{\mathbf{H}}_t, \bar{\mathbf{Z}}_{t,\tau+1}, \bar{\mathbf{a}}_{t,\tau}) \times p_\theta(\mathbf{Z}_t \mid \bar{\mathbf{H}}_t) \times$$
$$\Pi_{s=t+1}^{t+\tau} p_\theta(\mathbf{Z}_s | \mathbf{Z}_{s-1}, \mathbf{a}_{s-1}) d\bar{\mathbf{Z}}_{t,\tau+1}. \quad (7)$$

For the complete proof and detailed decomposition of the ELBO terms, we refer readers to the Appendix.

### 4.2. Model Architecture

**Generative model.** Our causal model in Figure 2 assumes the following factorization of the generative model:

$$p_\theta(\mathbf{Y}_{t+\tau}, \bar{\mathbf{Z}}_{t,\tau+1} | \bar{\mathbf{H}}_t, \bar{\mathbf{a}}_{t,\tau}) = p_\theta(\mathbf{Y}_{t+\tau} | \mathbf{Z}_{t+\tau})$$
$$\times \Pi_{s=t}^{t+\tau-1} p_\theta(\mathbf{Z}_{s+1} | \mathbf{Z}_s, \bar{\mathbf{a}}_{s,\tau_s}) \times p_\theta(\mathbf{Z}_t | \bar{\mathbf{H}}_t), \quad (8)$$

where $\tau_s = \tau + t - s$ and each conditional distribution is parameterized as a Gaussian with neural network-derived parameters:

$$p_\theta(\mathbf{Y}_{t+\tau} \mid \cdot) = \mathcal{N}(\mu_y^\theta(\mathbf{Z}_{t+\tau}), \sigma_y^{\theta\,2}(\mathbf{Z}_{t+\tau})), \quad (9)$$
$$p_\theta(\mathbf{Z}_{s+1} \mid \cdot) = \mathcal{N}(\mu_{z_1}^\theta(\mathbf{Z}_s, \bar{\mathbf{a}}_{s,\tau_s}), \sigma_{z_1}^{\theta\,2}(\mathbf{Z}_s, \bar{\mathbf{a}}_{s,\tau_s})), \quad (10)$$
$$p_\theta(\mathbf{Z}_t \mid \cdot) = \mathcal{N}(\mu_{z_0}^\theta(\bar{\mathbf{H}}_t), \sigma_{z_0}^{\theta\,2}(\bar{\mathbf{H}}_t)). \quad (11)$$

Specifically, we employ a multilayer perceptron (MLP) for $\mu_y$ and $\sigma_y$ to decode the final system state $\mathbf{Z}_{t+\tau}$ into the target output $\mathbf{Y}_{t+\tau}$. Given the sequential nature of historical information $\bar{\mathbf{H}}_t$, we utilize Long Short-Term Memory (LSTM) (Hochreiter & Schmidhuber, 1997) architectures for $\mu_{z_0}$ and $\sigma_{z_0}$ to encode the initial system state $\mathbf{Z}_t$. Similarly, the state transition functions $\mu_{z_1}$ and $\sigma_{z_1}$, which map the current state and intervention to the next state, are also implemented using LSTMs to capture the iterative dynamics of the system.

For the intervention distributions in Equation 6, we model them differently based on the type of interventions. When

**Algorithm 1** Optimize Intervention Sequence

---

**Require:** Historical information $\bar{\mathbf{H}}_t$, target $\mathbf{Y}_{\text{target}}$, trained VCIP model, learning rate $\alpha$, maximum epochs $M$
**Ensure:** Optimal intervention sequence $\bar{\mathbf{a}}_{t,\tau}^*$
1: Initialize $\bar{\mathbf{a}}_{t,\tau}$ randomly
2: **for** epoch = 1 to $M$ **do**
3:      Calculate $\mathcal{L}_{\text{ELBO}}$ using Equation 19
4:      $\bar{\mathbf{a}}_{t,\tau} \leftarrow \bar{\mathbf{a}}_{t,\tau} - \alpha \nabla_{\bar{\mathbf{a}}_{t,\tau}} \mathcal{L}_{\text{ELBO}}$
5: **end for**
6: **return** $\bar{\mathbf{a}}_{t,\tau}^* = \bar{\mathbf{a}}_{t,\tau}$

---

the interventions are continuous with values bounded in $[0, 1]$, we use Beta distributions:

$$p_\theta(\mathbf{a}_s \mid \cdot) = \mathrm{Beta}(\alpha_0^\theta(\mathbf{Z}_s), \beta_0^\theta(\mathbf{Z}_s)), \quad (12)$$
$$p_\theta(\bar{\mathbf{a}}_{t,\tau} \mid \cdot) = \Pi_{i=0}^{\tau-1} \mathrm{Beta}(\alpha_1^\theta(\bar{\mathbf{H}}_t, \bar{\mathbf{a}}_{t,i}), \beta_1^\theta(\bar{\mathbf{H}}_t, \bar{\mathbf{a}}_{t,i})). \quad (13)$$

When the interventions are binary, we use Bernoulli distributions:

$$p_\theta(\mathbf{a}_s \mid \cdot) = \mathrm{Bern}(\pi_0^\theta(\mathbf{Z}_s)), \quad (14)$$
$$p_\theta(\bar{\mathbf{a}}_{t,\tau} \mid \cdot) = \Pi_{i=0}^{\tau-1} \mathrm{Bern}(\pi_1^\theta(\bar{\mathbf{H}}_t, \bar{\mathbf{a}}_{t,i})). \quad (15)$$

where $\alpha_0$, $\beta_0$, and $\pi_0$ are implemented using MLPs, while $\alpha_1$, $\beta_1$, and $\pi_1$ are implemented using LSTM networks to capture the temporal dependencies in interventions.

**Inference model**. We use an inference model $q_\phi(\bar{\mathbf{Z}}_{t:t+\tau} \mid \bar{\mathbf{H}}_t, \mathbf{Y}_{t+\tau})$ to approximate the posterior distribution of the latent variables, which can be factorized as:

$$q_\phi(\bar{\mathbf{Z}}_{t:t+\tau} \mid \bar{\mathbf{H}}_t, \mathbf{Y}_{t+\tau}, \bar{\mathbf{a}}_{t,\tau}) = q_\phi(\mathbf{Z}_t \mid \bar{\mathbf{H}}_t)$$
$$\times \Pi_{s=t}^{t+\tau-1} q_\phi(\mathbf{Z}_{s+1} \mid \mathbf{Y}_{t+\tau}, \bar{\mathbf{a}}_{s,\tau_s}, \mathbf{Z}_s), \quad (16)$$

where $\tau_s = \tau + t - s$. Each of the corresponding factors is described as:

$$q_\phi(\mathbf{Z}_t | \cdot) = \mathcal{N}(\mu_{z_0}^\phi(\bar{\mathbf{H}}_t), \sigma_{z_0}^{\phi\,2}(\bar{\mathbf{H}}_t)), \quad (17)$$
$$q_\phi(\mathbf{Z}_{s+1} | \cdot) = \mathcal{N}(\mu_{z_1}^\phi(\mathbf{Y}_{t+\tau}, \bar{\mathbf{a}}_{s,\tau_s}, \mathbf{Z}_s), \sigma_{z_1}^{\phi\,2}(\mathbf{Y}_{t+\tau}, \bar{\mathbf{a}}_{s,\tau_s}, \mathbf{Z}_s)), \quad (18)$$

where $\mu, \sigma$ are implemented by LSTM networks. Note that $\mathbf{Z}_s$ depends on the descendant latent factors after time $s$. To implement this in practice, we use an LSTM to learn the representation of intervention sequences $\bar{\mathbf{a}}_{s,\tau_s}$. Specifically, at each time step $s$, we encode the current action $\mathbf{a}_s$ and subsequent interventions through an LSTM network, producing the latent representation of the intervention sequence.

### 4.3. Traning and Inference

To solve the optimal sequence in Equation 1, we first train VCIP to maximize $\mathrm{ELBO}_1$ under observational data. Ac-

cording to the previous section, we define:

$$\mathcal{L}_{\text{ELBO}} = \sum_{s=t}^{t+\tau} \text{KL}\left(q_\phi(\mathbf{Z}_s|\cdot) \parallel p_\theta(\mathbf{Z}_s|\cdot)\right)$$
$$- \mathbb{E}_{q_\phi}[\log p_\theta(\mathbf{Y}_{t+\tau} \mid \mathbf{Z}_{t+\tau})]$$
$$+ \lambda \left( \sum_{s=t}^{t+\tau-1} \mathbb{E}_{q_\phi}[\log p_\theta(\mathbf{a}_s \mid \mathbf{Z}_s)] - \log p_\theta(\bar{\mathbf{a}}_{t,\tau} \mid \bar{\mathbf{H}}_t) \right) \quad (19)$$

The first term on the RHS minimizes the KL divergence to align the inference model $q_\phi(\mathbf{Z}_s|\cdot)$ with the generative model $p_\theta(\mathbf{Z}_s|\cdot)$; the second term maximizes the probability of achieving the target output $\mathbf{Y}_{t+\tau}$. According to Theorem 1, optimizing these two terms is equivalent to maximizing the observational ELBO$_2$. When $\lambda = 1$, the third term bridges the gap between interventional and observational log-likelihoods.

During VCIP training, we split observational data into historical and future $\tau$-step information to optimize parameters $\phi$ and $\theta$ by minimizing the evidence lower bound $\mathcal{L}_{\text{ELBO}}$. After VCIP training, we fix all model parameters and optimize the intervention sequence as learnable parameters. The specific process is shown in Algorithm 1 [1]. For counterfactual-estimation methods, we employ a similar optimization algorithm, but the objective is to minimize the distance between the predicted output and the target.

# 5. Experiments

In this section, we conduct comprehensive experiments to evaluate the performance of VCIP against several state-of-the-art baselines. Specifically, we validate our model on both simulated tumor and real-world healthcare datasets to demonstrate its effectiveness in intervention ranking and counterfactual inference.

**Baselines.** We compare our method against state-of-the-art models for counterfactual estimation over time: **RMSN** (Lim et al., 2018), **CRN** (Bica et al., 2020), **CT** (Melnychuk et al., 2022), and **ACTIN** (Wang et al., 2024). To ensure fair comparison, we perform hyperparameter tuning for all baselines (see details in Appendix.)

**Datasets.** The tumor dataset simulates lung cancer treatment dynamics through a pharmacokinetic-pharmacodynamic framework (Geng et al., 2017). This dataset serves as a benchmark for evaluating causal inference in sequential decision-making (Lim et al., 2018; Bica et al., 2020). A key advantage of this bio-mathematical model is its confounding parameter $\gamma$ that modulates treatment history's impact on subsequent intervention decisions. Higher values of $\gamma$ correspond to stronger historical confounding effects in treatment allocation. To enhance clinical

---

[1]Code available at: https://github.com/wangxin0126/VCIP-ICML

relevance, we extend the binary intervention space to a continuous domain, capturing the dose-dependent nature of chemotherapy and radiotherapy treatments. The detailed data generation protocol is presented in Appendix.

For real-world evaluation, we leverage the Medical Information Mart for Intensive Care III (MIMIC-III) database (Johnson et al., 2016), which contains de-identified electronic health records from intensive care unit patients. The target outcome variable is diastolic blood pressure under two concurrent interventions: vasopressor administration and mechanical ventilation support. The feature set comprises 25 time-varying physiological measurements and 3 static patient characteristics, following the preprocessing protocol established in recent works (Hatt & Feuerriegel, 2021; Kuzmanovic et al., 2021; Melnychuk et al., 2022).

## 5.1. Ranking-based Evaluation

To evaluate the model's ability in ranking intervention sequences, we design a ranking-based evaluation framework. Specifically, for each test sample $(\bar{\mathbf{H}}_t, \bar{\mathbf{a}}_{t,\tau}, \mathbf{Y}_{\text{target}})$, where $\mathbf{Y}_{\text{target}} = \mathbf{Y}[\bar{\mathbf{a}}_{t,\tau}]$ represents the output corresponding to the ground truth sequence, we randomly generate $k$ candidate sequences $\{\bar{\mathbf{a}}_{t,\tau}^{(i)}\}_{i=1}^k$, with some generated by adding random perturbations to the ground truth sequence. For details, please refer to Appendix D. This results in an evaluation set containing $k + 1$ intervention sequences including the ground truth sequence.

Based on this evaluation set, we obtain sequence rankings through different models. For baseline models capable of predicting conditional expectations $\hat{\mathbf{Y}}[\bar{\mathbf{a}}_{t,\tau}] = \mathbb{E}[\mathbf{Y}[\bar{\mathbf{a}}_{t,\tau}] \mid \bar{\mathbf{H}}_t]$, we rank the intervention sequences in ascending order based on the deviation between predicted and target values $\|\hat{\mathbf{Y}}[\bar{\mathbf{a}}_{t,\tau}^{(i)}] - \mathbf{Y}_{\text{target}}\|$, denoted as $\mathbf{r}_{\text{pred}}$. A higher ranking position indicates that the intervention sequence is more likely to achieve the target output $\mathbf{Y}_{\text{target}}$. For the VCIP model, we rank sequences based on the ELBO loss $\mathcal{L}_{\text{ELBO}}$, where a lower value indicates a higher conditional likelihood $\log p_\theta(\mathbf{Y}[\bar{\mathbf{a}}_{t,\tau}^{(i)}] = \mathbf{Y}_{\text{target}} \mid \bar{\mathbf{H}}_t)$.

Based on the predicted ranking $\mathbf{r}_{\text{pred}}$, we propose the Ground Truth Ranking Position (GRP) metric to evaluate the normalized ranking of the ground truth sequence:

$$\text{GRP} = \frac{k + 1 - \xi}{k} \quad (20)$$

where $\xi$ denotes the position of the ground truth sequence in the predicted ranking (from 1 to $k + 1$). A higher GRP value indicates a better ranking position of the ground truth sequence, reflecting better model performance.

To comprehensively evaluate the model's ranking ability, we further propose the Ranking Correlation Score (RCS) metric. Specifically, we first obtain the true ranking $\mathbf{r}_{\text{true}}$

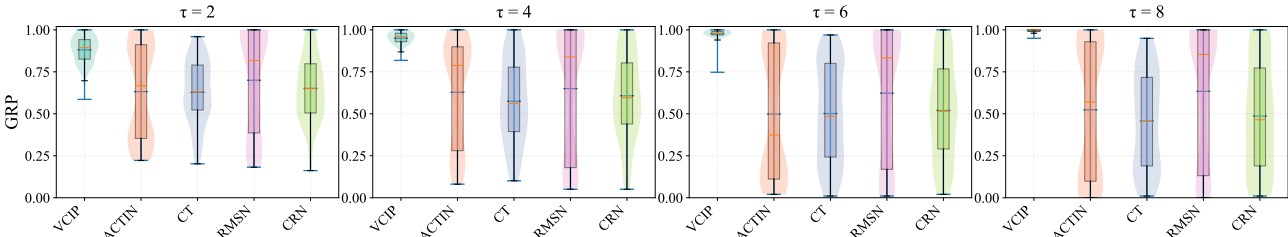

Figure 3. Comparison of GRP across different models on MIMIC-III dataset with varying prediction horizons ($\tau = 2, 4, 6, 8$). Higher GRP indicates better ranking performance.

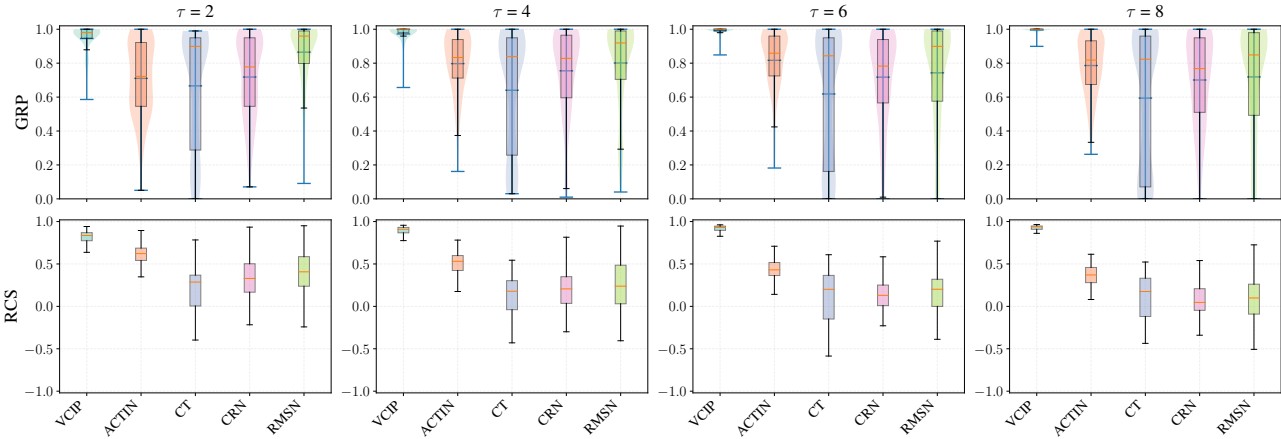

Figure 4. Evaluation results on the tumor simulation dataset with varying prediction horizons ($\tau = 2, 4, 6, 8$) and fixed confounding bias level ($\gamma = 4$). The upper row shows GRP and the lower row presents the RCS that measures the correlation between predicted and ground truth rankings (ranging from -1 to 1), where higher values indicate better performance in both metrics.

by sorting all sequences in ascending order based on their actual target distances $\|\mathbf{Y}[\bar{\mathbf{a}}_{t,\tau}^{(i)}] - \mathbf{Y}_{\text{target}}\|$. RCS is defined as the Spearman correlation coefficient between the predicted ranking $\mathbf{r}_{\text{pred}}$ and the true ranking $\mathbf{r}_{\text{true}}$ (Spearman, 1961):

$$\text{RCS} = \rho(\mathbf{r}_{\text{pred}}, \mathbf{r}_{\text{true}}) \tag{21}$$

where $\rho(\cdot, \cdot)$ denotes the Spearman correlation coefficient. The RCS value ranges from $[-1, 1]$, with values closer to 1 indicating better alignment between the predicted and true rankings, reflecting more accurate assessment of intervention sequence quality by the model.

**Results.** We first evaluate different models' GRP performance on the real-world MIMIC-III dataset, with results shown in Figure 3. Overall, our proposed VCIP model significantly outperforms baseline methods across all prediction horizons $\tau$, demonstrating its superiority in ranking intervention sequences. From the perspective of prediction horizon, baseline methods based on counterfactual estimation (ACTIN, CT, RMSN, and CRN) tend to suffer from performance degradation with larger $\tau$, due to the accumulation

of prediction errors. In contrast, VCIP, which directly computes ELBO using target outputs and intervention sequences, not only maintains but improves its ranking accuracy with larger $\tau$. This improvement may be attributed to the model's ability to leverage richer intervention information contained in longer sequences.

On the simulated tumor dataset with fixed confounding bias ($\gamma = 4$), we compare the GRP and RCS metrics across different models, with results shown in Figure 4. The evaluation demonstrates that VCIP significantly outperforms baseline models on both GRP and RCS metrics across all prediction horizons ($\tau = 2, 4, 6, 8$). Notably, the mean RCS maintains a high level above 0.7, indicating that VCIP's predicted intervention sequence rankings strongly correlate with ground truth rankings and accurately reflect how different intervention sequences approach the target state. Furthermore, as the prediction horizon increases, VCIP exhibits an increasing performance advantage over baseline models, demonstrating robust predictive stability.

*Table 1.* The long-range prediction results on the tumor dataset ($\gamma = 4$) under identical intervention strategies for both training and test sets are reported as target distance (mean $\pm$ std over five runs).

| | $\tau = 1$ | $\tau = 2$ | $\tau = 4$ | $\tau = 6$ | $\tau = 8$ | $\tau = 9$ | $\tau = 10$ | $\tau = 11$ | $\tau = 12$ |
|---|---|---|---|---|---|---|---|---|---|
| ACTIN | $0.42 \pm 0.14$ | $0.71 \pm 0.13$ | $1.05 \pm 0.21$ | $1.30 \pm 0.13$ | $1.47 \pm 0.13$ | $1.55 \pm 0.13$ | $1.59 \pm 0.16$ | $1.63 \pm 0.18$ | $1.68 \pm 0.21$ |
| CT | $0.55 \pm 0.20$ | $0.88 \pm 0.25$ | $1.43 \pm 0.36$ | $1.69 \pm 0.27$ | $1.87 \pm 0.39$ | $2.01 \pm 0.43$ | $2.04 \pm 0.46$ | $2.10 \pm 0.55$ | $2.14 \pm 0.59$ |
| CRN | $0.38 \pm 0.09$ | $0.60 \pm 0.08$ | $0.92 \pm 0.08$ | $1.19 \pm 0.18$ | $1.33 \pm 0.21$ | $1.40 \pm 0.24$ | $1.49 \pm 0.32$ | $1.59 \pm 0.41$ | $1.62 \pm 0.47$ |
| RMSN | $0.30 \pm 0.10$ | $0.45 \pm 0.10$ | $0.75 \pm 0.16$ | $0.98 \pm 0.22$ | $1.15 \pm 0.23$ | $1.22 \pm 0.25$ | $1.28 \pm 0.34$ | $1.43 \pm 0.32$ | $1.47 \pm 0.35$ |
| VCIP | $\mathbf{0.29 \pm 0.08}$ | $\mathbf{0.42 \pm 0.13}$ | $\mathbf{0.60 \pm 0.15}$ | $\mathbf{0.75 \pm 0.20}$ | $\mathbf{0.92 \pm 0.24}$ | $\mathbf{0.95 \pm 0.24}$ | $\mathbf{0.99 \pm 0.27}$ | $\mathbf{1.04 \pm 0.27}$ | $\mathbf{1.09 \pm 0.29}$ |

*Table 2.* The long-range prediction results on the tumor dataset ($\gamma = 4$) with distinct intervention strategies applied to training and test sets are reported as target distance (mean $\pm$ std over five runs).

| | $\tau = 1$ | $\tau = 2$ | $\tau = 4$ | $\tau = 6$ | $\tau = 8$ | $\tau = 9$ | $\tau = 10$ | $\tau = 11$ | $\tau = 12$ |
|---|---|---|---|---|---|---|---|---|---|
| ACTIN | $0.45 \pm 0.13$ | $0.78 \pm 0.09$ | $1.15 \pm 0.08$ | $1.49 \pm 0.07$ | $1.77 \pm 0.23$ | $1.90 \pm 0.25$ | $1.94 \pm 0.29$ | $2.06 \pm 0.27$ | $2.10 \pm 0.30$ |
| CT | $0.62 \pm 0.19$ | $0.93 \pm 0.21$ | $1.56 \pm 0.25$ | $1.98 \pm 0.28$ | $2.21 \pm 0.37$ | $2.34 \pm 0.43$ | $2.33 \pm 0.45$ | $2.40 \pm 0.40$ | $2.38 \pm 0.40$ |
| CRN | $0.47 \pm 0.11$ | $0.69 \pm 0.10$ | $1.03 \pm 0.16$ | $1.30 \pm 0.20$ | $1.64 \pm 0.34$ | $1.73 \pm 0.36$ | $1.80 \pm 0.24$ | $1.91 \pm 0.29$ | $2.05 \pm 0.31$ |
| RMSN | $0.37 \pm 0.11$ | $0.53 \pm 0.07$ | $0.81 \pm 0.14$ | $1.04 \pm 0.24$ | $1.34 \pm 0.30$ | $1.48 \pm 0.31$ | $1.59 \pm 0.22$ | $1.72 \pm 0.28$ | $1.83 \pm 0.22$ |
| VCIP | $\mathbf{0.29 \pm 0.04}$ | $\mathbf{0.44 \pm 0.06}$ | $\mathbf{0.71 \pm 0.09}$ | $\mathbf{0.87 \pm 0.10}$ | $\mathbf{1.08 \pm 0.22}$ | $\mathbf{1.24 \pm 0.26}$ | $\mathbf{1.28 \pm 0.25}$ | $\mathbf{1.30 \pm 0.18}$ | $\mathbf{1.36 \pm 0.26}$ |

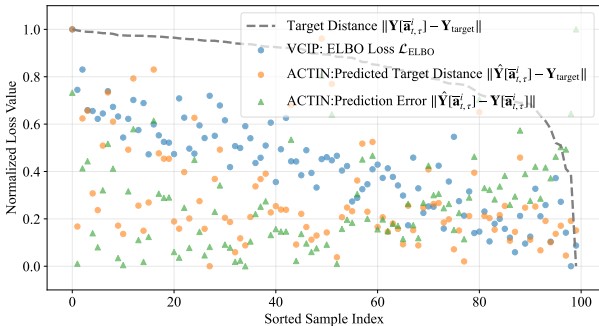

*Figure 5.* Individual-level predictions under 100 candidate interventions on the tumor dataset ($\tau = 6$, $\gamma = 4$) for VCIP and ACTIN. Each point represents a normalized prediction score for a specific intervention option.

**Case study.** To investigate why baselines perform poorly in RCS ranking, we conduct a case study by selecting representative samples. As shown in Figure 5, the experimental results reveal a fundamental limitation of counterfactual methods based on predicted potential outcomes for ranking. Specifically, there exists a significant discrepancy between the predicted target distance $\|\hat{\mathbf{Y}}[\bar{\mathbf{a}}^i_{t,\tau}] - \mathbf{Y}_{\text{target}}\|$ and the true target distance $\|\mathbf{Y}[\bar{\mathbf{a}}^i_{t,\tau}] - \mathbf{Y}_{\text{target}}\|$ (grey dashed line). This discrepancy stems from the prediction error $\|\hat{\mathbf{Y}}[\bar{\mathbf{a}}^i_{t,\tau}] - \mathbf{Y}[\bar{\mathbf{a}}^i_{t,\tau}]\|$, which cannot be evaluated in practice due to the unobservability of the counterfactual outcome $\mathbf{Y}[\bar{\mathbf{a}}^i_{t,\tau}]$. In contrast, VCIP's ELBO loss better preserves the monotonic relationship with the true target distance, resulting in more reliable ranking.

## 5.2. Intervention Sequence Optimization

When interventions are continuous-valued, the approach of selecting from a finite set of ranked intervention candidates may prove inadequate for capturing the full spectrum of possible interventions. To address this limitation, we propose gradient-based intervention sequence optimization algorithms for both the baseline methods and VCIP. In this section, we evaluate the models' capability to optimize intervention sequences for achieving desired outcomes under continuous intervention scenarios. Specifically, given a target trajectory $\mathbf{Y}_{\text{target}}$ for an individual, we first obtain the optimal intervention sequence $\bar{\mathbf{a}}^*_{t,\tau}$ using Algorithm 1, then compute the target distance $\|\mathbf{Y}[\bar{\mathbf{a}}^*_{t,\tau}] - \mathbf{Y}_{\text{target}}\|$ between the potential outcome and the target trajectory. A smaller distance indicates that the obtained optimal sequence leads to potential outcomes closer to the target output.

**Results.** Due to the unobservable nature of counterfactuals, we conduct evaluations exclusively on the simulated tumor dataset. We compare the performance of different models across varying confounding levels ($\gamma = 1, 2, 3$) and prediction horizons ($\tau = 1$ to $\tau = 6$), with results illustrated in Figure 6. The experimental results demonstrate that VCIP achieves comparable performance when confounding is minimal ($\gamma = 1$) and shows superior performance as confounding intensities increase. While the prediction errors of all models increase with both the confounding strength and prediction horizon, VCIP maintains a clear advantage over baseline methods under stronger confounding (especially for $\tau = 2, 3$), exhibiting better stability and lower prediction errors in long-range predictions.

To further investigate the impact of prediction horizon on model performance, we present the results for different prediction horizons ($\tau = 1$ to $\tau = 12$) with $\gamma = 4$ in Table 1.

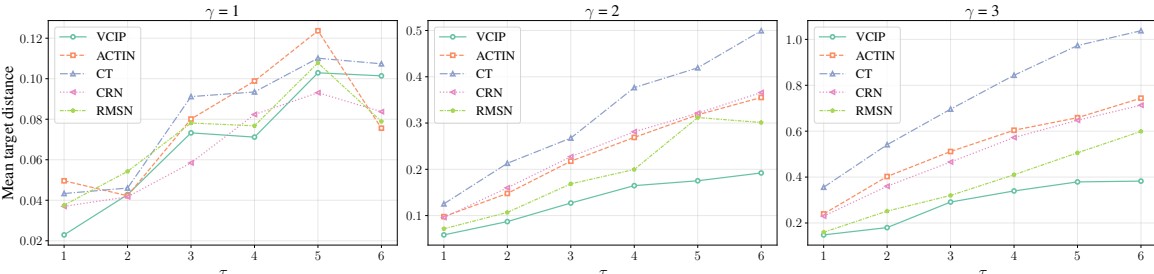

*Figure 6.* Comparison of target distances across different models on the tumor dataset under varying confounding levels. The x-axis represents different time steps, and the y-axis shows the mean target distance. Each subplot corresponds to a different confounding levels.

*Table 3.* Ablation study results comparing model performance with and without confounding adjustment.

| | GPR | | RCS | | Target distance | | | |
| --- | --- | --- | --- | --- | --- | --- | --- | --- |
| | $\tau = 2$ | $\tau = 4$ | $\tau = 2$ | $\tau = 4$ | $\gamma = 1$ | $\gamma = 2$ | $\gamma = 3$ | $\gamma = 4$ |
| RMSN | **0.863 ± 0.181** | **0.796 ± 0.267** | 0.400 ± 0.311 | **0.251 ± 0.299** | **0.078 ± 0.017** | 0.301 ± 0.131 | **0.599 ± 0.204** | **0.985 ± 0.215** |
| RMSN w/o adjustment | 0.797 ± 0.318 | 0.747 ± 0.348 | **0.461 ± 0.460** | 0.213 ± 0.407 | 0.087 ± 0.020 | **0.280 ± 0.144** | 0.694 ± 0.201 | 1.132 ± 0.090 |
| VCIP | **0.944 ± 0.091** | **0.972 ± 0.063** | **0.772 ± 0.206** | **0.869 ± 0.156** | 0.101 ± 0.069 | **0.192 ± 0.101** | **0.382 ± 0.085** | **0.746 ± 0.198** |
| VCIP w/o adjustment | 0.791 ± 0.334 | 0.796 ± 0.356 | 0.566 ± 0.508 | 0.595 ± 0.518 | **0.092 ± 0.080** | 0.284 ± 0.134 | 0.756 ± 0.299 | 0.912 ± 0.194 |

As shown in the table, the target distance increases with longer prediction horizons across all models. VCIP consistently achieves the lowest target distance across all prediction horizons, outperforming all baselines by a substantial margin. Specifically, for short-term predictions ($\tau = 1$), VCIP achieves a target distance of 0.29, showing a modest improvement of 3.3% compared to RMSN (0.30). The performance gap widens significantly for long-term predictions. When $\tau = 12$, VCIP maintains a target distance of 1.09, outperforming the best baseline RMSN (1.47) by 25.9%, demonstrating VCIP's superior capability in handling long-range predictions under strong confounding ($\gamma = 4$).

When applying identical intervention strategies for both training and test sets, VCIP demonstrates superior performance as shown above. To further evaluate the generalization capability of different models under more challenging scenarios, we conduct experiments where the intervention strategies differ between training and test sets (see Appendix for details of strategy generation). The results are presented in Table 2. Even under this more challenging setting, VCIP maintains its superior performance across all prediction horizons. For short-term predictions ($\tau = 1$), VCIP achieves a target distance of 0.29, outperforming the best baseline RMSN (0.37) by 21.6%. The advantage becomes more pronounced for long-term predictions, where at $\tau = 12$, VCIP achieves a target distance of 1.36, showing a significant improvement of 25.7% compared to RMSN (1.83). These results show VCIP's strong generalization capability when facing unseen intervention strategies.

### 5.3. Abalation Study

Previous research has shown that addressing confounding bias through proper adjustment is crucial for accurate counterfactual estimation in longitudinal settings. In this work, we employ g-formula for adjustment, specifically using the last term in Equation (19) to bridge interventional and observational log-likelihoods. To evaluate the effectiveness of adjustment, we conduct ablation studies by setting $\lambda = 0$ in Equation (19) and design corresponding ablations for RMSN. Table 3 presents the results for both ranking tasks (at $\gamma = 4$) and sequence optimization tasks (with $\tau = 6$ evaluated across different $\gamma$ values).

The ranking results demonstrate that adjustment significantly enhances the performance of both models. For VCIP, incorporating adjustment improves the GPR metric to 0.944 and 0.972 at $\tau = 2$ and $\tau = 4$ respectively (compared to 0.791 and 0.796 without adjustment), while RCS shows stronger ranking capability (0.772 and 0.869 vs. 0.566 and 0.595). Similarly, RMSN's GPR improves from 0.797 and 0.747 to 0.863 and 0.796 at respective horizons, accompanied by enhanced RCS scores. These results indicate that adjustment enables more accurate evaluation and ranking of intervention sequences.

From the target distance perspective, adjustment helps both models achieve more stable performance under strong confounding. Notably, at high confounding levels ($\gamma = 3, 4$), VCIP with adjustment achieves target distances of 0.382 and 0.746, substantially outperforming its counterpart without adjustment (0.756 and 0.912). RMSN exhibits a similar trend, with adjustment yielding better results at $\gamma = 4$

(0.985 vs. 1.132). This confirms that adjustment effectively handles confounding effects, enabling the models to identify superior intervention sequences. Interestingly, the performance gains from adjustment are less pronounced under low confounding conditions ($\gamma = 1, 2$).

## 6. Conclusion

This paper introduces the novel counterfactual target achievement problem, which aims to identify intervention sequences that guide systems toward desired target outcomes, addressing a critical need in many real-world applications like personalized healthcare. To address this problem, we propose the VCIP framework that models the likelihood of achieving target states through intervention sequences. Unlike traditional counterfactual estimation methods that accumulate prediction errors, VCIP provides a more robust solution by learning target achievement probability. Extensive experiments on both synthetic and real-world datasets validate VCIP's effectiveness.

## Acknowledgements

This research is supported by National Key R&D Program of China (No. 2021ZD0111700), National Nature Science Foundation of China (No. 62137002, 62176245), Special Foundation for Science and Technology Innovation and Entrepreneurship of CCTEG NO. 2020-2-TD-CXY006, Key Science and Technology Special Project of Anhui Province (No. 202103a07020002). We thank the anonymous reviewers for their constructive comments that help improve the manuscript.

## Impact Statement

This research advances temporal causal inference by introducing a framework for identifying intervention sequences that guide systems toward specific target outcomes, with applications in healthcare and economic policy-making. While our method can help practitioners make more informed decisions about sequential interventions to achieve desired targets (e.g., normalizing physiological indicators or stabilizing economic metrics), we emphasize that it should serve as a decision support tool rather than replace expert judgment. The framework's effectiveness depends on training data quality and representativeness, and careful consideration must be given to potential biases and the need for ongoing validation before deployment in critical real-world applications.

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

## A. Assumptions

For accurate estimation of treatment effects using observational data, we consider three fundamental assumptions that form the theoretical foundation of our work. These assumptions align with established frameworks in causal inference literature (Lim et al., 2018; Bica et al., 2020; Li et al., 2021; Melnychuk et al., 2022; Wang et al., 2024):

**Assumption A.1 (Consistency).** The observed outcome $\mathbf{Y}_{t+1}$ matches the potential outcome $\mathbf{Y}_{t+1}[\mathbf{a}_t]$ when treatment $\mathbf{a}_t$ is applied at time $t$, expressed as $\mathbf{Y}_{t+1} = \mathbf{Y}_{t+1}[\mathbf{a}_t]$.

**Assumption A.2 (Sequential Overlap).** For any treatment $\mathbf{a}_t$ at time $t$, there exists a non-zero probability of assignment conditioned on the historical trajectory $\bar{\mathbf{h}}_t$. Mathematically, $0 < P(\mathbf{A}_t = \mathbf{a}_t \mid \bar{\mathbf{H}}_t = \bar{\mathbf{h}}_t) < 1$, $\forall \mathbf{a}_t \in \mathcal{A}$ when $P(\bar{\mathbf{H}}_t = \bar{\mathbf{h}}_t) > 0$.

**Assumption A.3 (Sequential Ignorability).** Treatment assignment at time $t$ exhibits conditional independence from potential outcomes at $t + 1$, given the observed history: $\mathbf{A}_t \perp \mathbf{Y}_{t+1}[\mathbf{a}_t] \mid \bar{\mathbf{H}}_t$, $\forall \mathbf{a}_t \in \mathcal{A}$. This ensures no unmeasured confounding affects the treatment-outcome relationship.

## B. Proof

We first derive the Evidence Lower BOund (ELBO) for both the log interventional likelihood and log observational likelihood.

For the log interventional likelihood:

$$
\begin{aligned}
&\log p_\theta(\mathbf{Y}[\bar{\mathbf{a}}_{t,\tau}] = \mathbf{Y}_{\text{target}} \mid \bar{\mathbf{H}}_t) \\
&= \log \int p_\theta(\mathbf{Y}_{\text{target}} \mid \bar{\mathbf{H}}_t, \bar{\mathbf{a}}_{t,\tau}, \bar{\mathbf{Z}}_{t,\tau+1}) \, p_\theta(\mathbf{Z}_t \mid \bar{\mathbf{H}}_t) \, \Pi_{s=t+1}^{t+\tau} p_\theta(\mathbf{Z}_s \mid \mathbf{Z}_{s-1}, \mathbf{a}_{s-1}) d\bar{\mathbf{Z}}_{t,\tau+1} \\
&= \log \int q_\phi(\bar{\mathbf{Z}}_{t,\tau+1} \mid \bar{\mathbf{H}}_t, \mathbf{Y}_{\text{target}}, \bar{\mathbf{a}}_{t,\tau}) \frac{p_\theta(\mathbf{Y}_{\text{target}}, \bar{\mathbf{Z}}_{t,\tau+1} \mid \bar{\mathbf{H}}_t, do(\bar{\mathbf{a}}_{t,\tau}))}{q_\phi(\bar{\mathbf{Z}}_{t,\tau+1} \mid \bar{\mathbf{H}}_t, \mathbf{Y}_{\text{target}}, \bar{\mathbf{a}}_{t,\tau})} d\bar{\mathbf{Z}}_{t,\tau+1} \\
&\geq \mathbb{E}_{q_\phi}[\log \frac{p_\theta(\mathbf{Y}_{\text{target}}, \bar{\mathbf{Z}}_{t,\tau+1} \mid \bar{\mathbf{H}}_t, do(\bar{\mathbf{a}}_{t,\tau}))}{q_\phi(\bar{\mathbf{Z}}_{t,\tau+1} \mid \bar{\mathbf{H}}_t, \mathbf{Y}_{\text{target}}, \bar{\mathbf{a}}_{t,\tau})}] := \text{ELBO}_1
\end{aligned}
\tag{22}
$$

For the log observational likelihood, we derive two different formulations. The first formulation is:

$$
\begin{aligned}
&\log p_\theta(\mathbf{Y}_{t+\tau} = \mathbf{Y}_{\text{target}} \mid \bar{\mathbf{H}}_t, \bar{\mathbf{a}}_{t,\tau}) \\
&= \log \int p_\theta(\mathbf{Y}_{\text{target}}, \bar{\mathbf{Z}}_{t,\tau+1} \mid \bar{\mathbf{H}}_t, \bar{\mathbf{a}}_{t,\tau}) \, d\bar{\mathbf{Z}}_{t,\tau+1} \\
&= \log \int q_\phi(\bar{\mathbf{Z}}_{t,\tau+1} \mid \bar{\mathbf{H}}_t, \mathbf{Y}_{\text{target}}, \bar{\mathbf{a}}_{t,\tau}) \frac{p_\theta(\mathbf{Y}_{\text{target}}, \bar{\mathbf{Z}}_{t,\tau+1} \mid \bar{\mathbf{H}}_t, \bar{\mathbf{a}}_{t,\tau})}{q_\phi(\bar{\mathbf{Z}}_{t,\tau+1} \mid \bar{\mathbf{H}}_t, \mathbf{Y}_{\text{target}}, \bar{\mathbf{a}}_{t,\tau})} \, d\bar{\mathbf{Z}}_{t,\tau+1} \\
&\geq \mathbb{E}_{q_\phi} \left[ \log \frac{p_\theta(\mathbf{Y}_{\text{target}}, \bar{\mathbf{Z}}_{t,\tau+1} \mid \bar{\mathbf{H}}_t, \bar{\mathbf{a}}_{t,\tau})}{q_\phi(\bar{\mathbf{Z}}_{t,\tau+1} \mid \bar{\mathbf{H}}_t, \mathbf{Y}_{\text{target}}, \bar{\mathbf{a}}_{t,\tau})} \right] := \text{ELBO}_2
\end{aligned}
\tag{23}
$$

The alternative formulation of the log observational likelihood is:

$$\log p_\theta(\mathbf{Y}_{t+\tau} = \mathbf{Y}_{\text{target}} | \bar{\mathbf{H}}_t, \bar{\mathbf{a}}_{t,\tau})$$

$$= \log p_\theta(\mathbf{Y}_{\text{target}}, \bar{\mathbf{H}}_t, \bar{\mathbf{a}}_{t,\tau}) - \log p_\theta(\bar{\mathbf{H}}_t, \bar{\mathbf{a}}_{t,\tau})$$

$$= \log \int p_\theta(\mathbf{Y}_{\text{target}}, \bar{\mathbf{H}}_t, \bar{\mathbf{a}}_{t,\tau}, \bar{\mathbf{Z}}_{t,\tau+1}) d\bar{\mathbf{Z}}_{t,\tau+1} - \log p_\theta(\bar{\mathbf{H}}_t, \bar{\mathbf{a}}_{t,\tau})$$

$$= \log \int p_\theta(\mathbf{Y}_{\text{target}} \mid \bar{\mathbf{H}}_t, \bar{\mathbf{a}}_{t,\tau}, \bar{\mathbf{Z}}_{t,\tau+1}) p_\theta(\bar{\mathbf{H}}_t, \bar{\mathbf{a}}_{t,\tau}, \bar{\mathbf{Z}}_{t,\tau+1}) d\bar{\mathbf{Z}}_{t,\tau+1} - \log p_\theta(\bar{\mathbf{H}}_t, \bar{\mathbf{a}}_{t,\tau})$$

$$= \log \int p_\theta(\mathbf{Y}_{\text{target}} \mid \bar{\mathbf{H}}_t, \bar{\mathbf{a}}_{t,\tau}, \bar{\mathbf{Z}}_{t,\tau+1}) \, p_\theta(\bar{\mathbf{H}}_t) p_\theta(\mathbf{Z}_t \mid \bar{\mathbf{H}}_t) \, \Pi_{s=t+1}^{t+\tau} \, p_\theta(\mathbf{Z}_s \mid \mathbf{Z}_{s-1}, \mathbf{a}_{s-1}) \, \Pi_{s=t}^{t+\tau} \, p_\theta(\mathbf{a}_s \mid \mathbf{Z}_s) d\bar{\mathbf{Z}}_{t,\tau+1}$$

$$- \log p_\theta(\bar{\mathbf{H}}_t, \bar{\mathbf{a}}_{t,\tau})$$

$$= \log \int p_\theta(\mathbf{Y}_{\text{target}}, \bar{\mathbf{Z}}_{t,\tau+1} | \bar{\mathbf{H}}_t, do(\bar{\mathbf{a}}_{t,\tau})) \, p_\theta(\bar{\mathbf{H}}_t) \, \Pi_{s=t}^{t+\tau} \, p_\theta(\mathbf{a}_s \mid \mathbf{Z}_s) d\bar{\mathbf{Z}}_{t,\tau+1} - \log p_\theta(\bar{\mathbf{H}}_t, \bar{\mathbf{a}}_{t,\tau})$$

$$= \log \int q_\phi(\bar{\mathbf{Z}}_{t,\tau+1} \mid \bar{\mathbf{H}}_t, \mathbf{Y}_{\text{target}}, \bar{\mathbf{a}}_{t,\tau}) \frac{p_\theta(\mathbf{Y}_{\text{target}}, \bar{\mathbf{Z}}_{t,\tau+1} | \bar{\mathbf{H}}_t, do(\bar{\mathbf{a}}_{t,\tau}))}{q_\phi(\bar{\mathbf{Z}}_{t,\tau+1} \mid \bar{\mathbf{H}}_t, \mathbf{Y}_{\text{target}}, \bar{\mathbf{a}}_{t,\tau})} \, p_\theta(\bar{\mathbf{H}}_t) \, \Pi_{s=t}^{t+\tau} \, p_\theta(\mathbf{a}_s \mid \mathbf{Z}_s) d\bar{\mathbf{Z}}_{t,\tau+1} - \log p_\theta(\bar{\mathbf{H}}_t, \bar{\mathbf{a}}_{t,\tau})$$

$$\geq \mathbb{E}_{q_\phi}[\log \frac{p_\theta(\mathbf{Y}_{\text{target}}, \bar{\mathbf{Z}}_{t,\tau+1} | \bar{\mathbf{H}}_t, do(\bar{\mathbf{a}}_{t,\tau}))}{q_\phi(\bar{\mathbf{Z}}_{t,\tau+1} \mid \bar{\mathbf{H}}_t, \mathbf{Y}_{\text{target}}, \bar{\mathbf{a}}_{t,\tau})} + \log p_\theta(\bar{\mathbf{H}}_t) + \sum_{s=t}^{t+\tau} \log p_\theta(\mathbf{a}_s \mid \mathbf{Z}_s)] - \log p_\theta(\bar{\mathbf{H}}_t, \bar{\mathbf{a}}_{t,\tau})$$

$$= \mathbb{E}_{q_\phi}[\log \frac{p_\theta(\mathbf{Y}_{\text{target}}, \bar{\mathbf{Z}}_{t,\tau+1} | \bar{\mathbf{H}}_t, do(\bar{\mathbf{a}}_{t,\tau}))}{q_\phi(\bar{\mathbf{Z}}_{t,\tau+1} \mid \bar{\mathbf{H}}_t, \mathbf{Y}_{\text{target}}, \bar{\mathbf{a}}_{t,\tau})} + \sum_{s=t}^{t+\tau} \log p_\theta(\mathbf{a}_s \mid \mathbf{Z}_s)] - \log p_\theta(\bar{\mathbf{a}}_{t,\tau} \mid \bar{\mathbf{H}}_t)$$

$$= \text{ELBO}_1 + \sum_{s=t}^{t+\tau} \mathbb{E}_{q_\phi}[\log p_\theta(\mathbf{a}_s \mid \mathbf{Z}_s)] - \log p_\theta(\bar{\mathbf{a}}_{t,\tau} \mid \bar{\mathbf{H}}_t) \tag{24}$$

The fifth equation follows from Property 1.

**Theorem B.1.** *Given the causal model illustrated in Figure 2, assume there exists constants $\epsilon_1, \epsilon_2 > 0$ such that:*

$$\mathcal{O} - \text{ELBO}_1 \leq \epsilon_1, \quad \mathcal{O}' - \text{ELBO}_2 \leq \epsilon_2.$$

*Then optimizing $\mathcal{O}$ can be approximated by optimizing:*

$$\text{ELBO}_2 - \sum_{s=t}^{t+\tau-1} \mathbb{E}_{q_\phi}[\log p_\theta(\mathbf{a}_s \mid \mathbf{Z}_s)] + \log p_\theta(\bar{\mathbf{a}}_{t,\tau} \mid \bar{\mathbf{H}}_t) \tag{25}$$

*with the error bounded by $\epsilon_1 + \epsilon_2$.*

*Proof.* From the alternative formulation of the observational distribution, we have:

$$\log p_\theta(\mathbf{Y}_{\text{target}} \mid \bar{\mathbf{H}}_t, \bar{\mathbf{a}}_{t,\tau}) \geq \text{ELBO}_1 + \sum_{s=t}^{t+\tau} \mathbb{E}_{q_\phi}[\log p_\theta(\mathbf{a}_s \mid \mathbf{Z}_s)] - \log p_\theta(\bar{\mathbf{a}}_{t,\tau} \mid \bar{\mathbf{H}}_t)$$

$$\log p_\theta(\mathbf{Y}_{\text{target}} \mid \bar{\mathbf{H}}_t, \bar{\mathbf{a}}_{t,\tau}) \geq \text{ELBO}_2$$

By the theorem assumptions:

$$\mathcal{O} - \text{ELBO}_1 \leq \epsilon_1, \quad \mathcal{O}' - \text{ELBO}_2 \leq \epsilon_2.$$

Substituting the first inequality into the first line of the observational distribution expression:

$$\mathcal{O} \leq \text{ELBO}_1 + \epsilon_1$$

$$\leq \log p_\theta(\mathbf{Y}_{\text{target}} \mid \bar{\mathbf{H}}_t, \bar{\mathbf{a}}_{t,\tau}) - \sum_{s=t}^{t+\tau} \mathbb{E}_{q_\phi}[\log p_\theta(\mathbf{a}_s \mid \mathbf{Z}_s)] + \log p_\theta(\bar{\mathbf{a}}_{t,\tau} \mid \bar{\mathbf{H}}_t) + \epsilon_1$$

$$\leq (\text{ELBO}_2 + \epsilon_2) - \sum_{s=t}^{t+\tau} \mathbb{E}_{q_\phi}[\log p_\theta(\mathbf{a}_s \mid \mathbf{Z}_s)] + \log p_\theta(\bar{\mathbf{a}}_{t,\tau} \mid \bar{\mathbf{H}}_t) + \epsilon_1$$

Therefore:

$$\mathcal{O} - (\text{ELBO}_2 - \sum_{s=t}^{t+\tau-1} \mathbb{E}_{q_\phi}[\log p_\theta(\mathbf{a}_s \mid \mathbf{Z}_s)] + \log p_\theta(\bar{\mathbf{a}}_{t,\tau} \mid \bar{\mathbf{H}}_t)) \leq \epsilon_1 + \epsilon_2$$

This establishes that optimizing the expression on the right-hand side approximates optimizing $\mathcal{O}$, with an error bounded by $\epsilon_1 + \epsilon_2$. □

## C. Problem Comparisons

To highlight the distinct characteristics of our proposed counterfactual target achievement problem, we compare it with two widely-adopted problems in sequential decision-making: counterfactual estimation and dynamic treatment regimes (DTRs). While all three approaches address temporal treatment planning, they differ fundamentally in their objectives, methodologies, and levels of analysis, as summarized in Table 4.

### C.1. Counterfactual Estimation

The counterfactual estimation framework focuses on predicting potential outcomes under specific treatment sequences. Given a patient's history $\bar{\mathbf{H}}_t = (\bar{\mathbf{X}}_t, \bar{\mathbf{A}}_{t-1}, \bar{\mathbf{Y}}_t, \mathbf{V})$, the goal is to estimate the expected outcome:

$$\mathbb{E}[\mathbf{Y}_{t+\tau}[\bar{\mathbf{a}}_{t,\tau}] | \bar{\mathbf{H}}_t] \tag{26}$$

This framework enables individual-level outcome prediction but does not directly optimize treatment sequences.

### C.2. Dynamic Treatment Regimes

DTRs provide a formal framework for personalizing treatments in sequential decision-making while optimizing the population-level outcomes. Unlike traditional fixed treatment protocols, DTRs adapt treatment decisions based on each patient's evolving characteristics and treatment history, yet aim to maximize the expected outcome across the entire patient population.

A DTR is defined as a structural causal model $\mathcal{M} = \langle \mathbf{U}, \mathbf{V}, \mathbf{F}, P(\mathbf{u}) \rangle$, where $\mathbf{U}$ represents unobserved exogenous variables, and $\mathbf{V} = \{\mathbf{X}_K, \mathbf{S}_K, Y\}$ comprises treatment variables $\mathbf{X}_K = \{X_1, \ldots, X_K\}$ over $K$ stages, time-varying covariates $\mathbf{S}_K = \{S_1, \ldots, S_K\}$, and a primary outcome $Y \in [0, 1]$. The structural functions $\mathbf{F}$ determine how variables evolve: state transitions follow $S_k \leftarrow \tau_k(\bar{x}_{k-1}, \bar{s}_{k-1}, \mathbf{u})$, treatment assignments are given by $X_k \leftarrow f_k(\bar{s}_k, \bar{x}_{k-1}, \mathbf{u})$, and the outcome is generated as $Y \leftarrow r(\bar{x}_K, \bar{s}_K, \mathbf{u})$. The distribution $P(\mathbf{u})$ specifies the underlying randomness in the system.

A treatment policy $\pi$ consists of a sequence of decision rules $\{\pi_k\}_{k=1}^K$, where each $\pi_k : \mathcal{S}_k \times \mathcal{X}_{k-1} \to \mathcal{P}(\mathcal{X}_k)$ maps the history of states and treatments to a distribution over the next treatment. When executed, the policy induces an interventional distribution:

$$P_\pi(\bar{x}_K, \bar{s}_K, y) = P_{\bar{x}_K}(y|\bar{s}_K) \prod_{k=0}^{K-1} P_{\bar{x}_k}(s_{k+1}|\bar{s}_k) \pi_{k+1}(x_{k+1}|\bar{s}_{k+1}, \bar{x}_k) \tag{27}$$

The core objective in DTR optimization is to find a policy $\pi^*$ that maximizes the expected outcome across the entire population:

$$\pi^* = \text{argmax}_{\pi \in \Pi} V_\pi(\mathcal{M}) = \text{argmax}_{\pi \in \Pi} \mathbb{E}_\pi[Y] \tag{28}$$

While DTRs have been extensively studied in the medical decision-making literature, our work addresses a fundamentally different problem. There are two key distinctions between DTRs and our proposed problem:

- **Optimization Objective:** DTRs aim to learn optimal treatment policies that maximize population-level outcomes ($\mathbb{E}_\pi[Y]$). In contrast, our problem focuses on identifying treatment sequences for specific individuals that maximize the probability of achieving a target state, formulated as $\bar{\mathbf{a}}^* = \text{argmax}_{\bar{\mathbf{a}}} p(\mathbf{Y}[\bar{\mathbf{a}}_{t,\tau}] = \mathbf{Y}_{\text{target}} \mid \bar{\mathbf{H}}_t)$.

*Table 4.* Comparison of key characteristics across different temporal treatment planning problems. Our counterfactual target achievement problem differs from existing approaches in its objective function, output form, analysis level, and decision timeline.

| Aspect | Counterfactual Target Achievement (ours) | Counterfactual Estimation | Dynamic Treatment Regimes |
|---|---|---|---|
| **Objective Function** | $\max_{\bar{\mathbf{a}}} p(\mathbf{Y}[\bar{\mathbf{a}}_{t,\tau}] = \mathbf{Y}_{\text{target}} \mid \bar{\mathbf{H}}_t)$ | $\mathbb{E}[\mathbf{Y}_{t+\tau}[\bar{\mathbf{a}}_{t,\tau}] \mid \bar{\mathbf{H}}_t]$ | $\max_{\pi \in \Pi} V_\pi(\mathcal{M}) = \max_{\pi \in \Pi} \mathbb{E}_\pi[Y]$ |
| **Output Form** | Complete treatment plan $\bar{\mathbf{a}}_{t,\tau}^*$ | Potential outcomes under treatment sequence $\bar{\mathbf{a}}_{t,\tau}$ | Decision rules $\{\pi_k : \mathcal{S}_k \times \mathcal{X}_{k-1} \to \mathcal{P}(\mathcal{X}_k)\}_{k=1}^K$ |
| **Analysis Level** | Individual trajectory optimization | Individual-level outcome prediction | Population-level optimization |
| **Decision Timeline** | Plan full sequence in advance | Estimate future outcomes given treatment sequence | Make decisions sequentially |

- **Analysis Level:** While DTR policies allow for individual-level adaptation, their optimization is fundamentally conducted at the population level. Our approach operates purely at the individual level, making predictions and optimizing treatment sequences based on each patient's unique historical trajectory $\bar{\mathbf{H}}_t$.

These fundamental differences in objectives and methodology explain why traditional DTR evaluation metrics and comparison frameworks are not directly applicable to our work. Our problem more closely aligns with personalized trajectory planning, where success is measured by the accuracy of individual-level predictions and the achievement of patient-specific target states.

## D. Datasets description

### D.1. Synthetic tumor dataset

Research presented in (Geng et al., 2017) employs a Tumor Growth (TG) simulator to forecast tumor volume progression over $t+1$ days following initial cancer detection, generating single-dimensional outputs. The framework encompasses two therapeutic interventions: radiation treatment ($\mathbf{A}_t^r$) and pharmaceutical therapy ($\mathbf{A}_t^c$). Our adaptation transforms the intervention variables from discrete to continuous values between 0 and 1. The therapeutic impacts manifest differently: radiation demonstrates immediate effects $d(t)$ on subsequent measurements, while pharmaceutical intervention exhibits extended influence $C(t)$ across multiple time points, as expressed by:

$$\mathbf{Y}_{t+1} = \left(1 + \rho \log\left(\frac{K}{\mathbf{Y}_t}\right) - \beta_C C(t) - (\alpha_r d(t) + \beta_r d(t)^2) + \epsilon_t\right) \mathbf{Y}_t, \tag{29}$$

The simulation parameters $\rho, K, \beta_C, \alpha_r, \beta_r$ are predefined, with $\epsilon_t$ representing random variation drawn from $N(0, 0.01^2)$. Patient-specific responses are characterized through parameters $\beta_C, \alpha_r, \beta_r$, derived from a three-component truncated normal mixture distribution. The mixture components serve as unchanging patient characteristics. Detailed parameter specifications are available in the anonymous repository. Time-dependent confounding affects both interventions through biased assignment protocols. The allocation probabilities for both treatments follow beta distributions:

$$\mathbf{A}_t^r, \mathbf{A}_t^c \sim \text{Beta}(2\sigma_t, 2 - 2\sigma_t), \tag{30}$$

where

$$\sigma_t = \sigma\left(\frac{\gamma}{D_{max}}\left(\bar{D}_{15}(\bar{\mathbf{Y}}_{t-1}) - D_{max}/2\right)\right), \tag{31}$$

Here, $\sigma(\cdot)$ denotes a sigmoid activation, $D_{max}$ represents maximum tumor size, $\bar{D}_{15}(\bar{\mathbf{Y}}_{t-1})$ indicates average tumor dimensions over a 15-day window, and $\gamma$ controls confounding intensity. When $\gamma$ equals zero, treatment assignments become purely random, while increasing values intensify time-varying confounding effects. In our implementation, $d_t$ and $C_t$ are modeled through cubic spline transformations of $\mathbf{A}_t^r$ and $\mathbf{A}_t^c$, respectively:

$$d(t) = 2\psi_r(\mathbf{A}_t^r), \tag{32}$$
$$C(t) = 5\psi_c(\mathbf{A}_t^c), \tag{33}$$

where $\psi_r$ and $\psi_c$ represent cubic spline transformations for the respective treatments. This nonlinear functional approach creates more realistic treatment-response relationships.

To ensure different strategies between testing and training phases, during the testing stage, interventions are randomly selected (with probability $\eta$) to follow an independent strategy, where $\mathbf{A}_t^r, \mathbf{A}_t^c \sim \text{Beta}(\alpha, \beta)$, with $\alpha$ and $\beta$ being constants independent of historical data.

The dataset comprises 1,000 training trajectories, 100 validation sequences, and 100 test cases across different confounding levels $\gamma$. Individual trajectories extend up to 60 time steps, though early termination may occur due to patient outcomes.

Following established methodologies (Bica et al., 2020; Melnychuk et al., 2022), performance evaluation utilizes normalized target distance, calculated relative to the maximum tumor volume $V_{\max} = 1150$ cubic centimeters.

### D.2. MIMIC-III Clinical Dataset

The Medical Information Mart for Intensive Care III (MIMIC-III) database (Johnson et al., 2016) constitutes a rich repository of de-identified clinical records from intensive care units. This comprehensive database encompasses diverse healthcare data elements including physiological measurements, medication administrations, laboratory results, clinical documentation, diagnostic classifications, and patient outcomes. To ensure systematic data processing and reproducibility, we employ the MIMIC-Extract framework (Wang et al., 2020), which implements standardized preprocessing procedures for the MIMIC-III database.

In alignment with recent methodological developments (Hatt & Feuerriegel, 2021; Kuzmanovic et al., 2021; Melnychuk et al., 2022), our analysis incorporates 25 longitudinal physiological indicators and 3 time-invariant patient characteristics. These features, particularly the categorical variables, undergo one-hot encoding transformation to facilitate numerical analysis. The intervention space comprises two binary treatment decisions commonly encountered in critical care settings: the administration of vasopressors and the implementation of mechanical ventilation support. We designate diastolic blood pressure as our primary outcome measure, given its clinical significance and sensitivity to both therapeutic interventions, thereby providing meaningful insights into patient trajectory management.

The experimental cohort consists of 5,000 patients with intensive care episodes spanning between 30 and 60 hours. This dataset undergoes a strategic partition into training (70%), validation (15%), and testing (15%) subsets. For temporal predictions extending $\tau$ steps ahead ($\tau \geq 2$), we establish $\tau_{\max}$ as an upper bound on the prediction horizon. We extract sub-trajectories of minimum length $\tau_{\max} + 1$ using rolling windows, with temporal alignment achieved by removing initial observations up to $\tau^{(i)} - \tau_{\max} + 1$, thus maintaining temporal causality. For immediate-horizon predictions ($\tau = 1$), we utilize complete trajectory sequences from the test cohort.

### D.3. Candidate Generation

We employ a hybrid approach to generate candidate sequences. Our framework creates random sequences (50%–80% of candidates) and perturbed ground truth sequences (20%–50% of candidates). The perturbation strategy is treatment-mode specific:

- **For discrete interventions:** We randomly flip bits in the ground truth sequence with probability 0.2.

- **For continuous interventions:** We apply context-aware shifts where values are modified based on their magnitude (low values shifted up, high values shifted down, middle values shifted randomly).

*Table 5.* Model comparison results across different interference levels ($\gamma$) and prediction horizons ($\tau$), with same intervention strategies applied during training and testing phases. Results are reported as target distance values (mean $\pm$ std over five runs).

| | | $\tau = 1$ | $\tau = 2$ | $\tau = 3$ | $\tau = 4$ | $\tau = 5$ | $\tau = 6$ |
|---|---|---|---|---|---|---|---|
| $\gamma = 1$ | ACTIN | 0.05±0.03 | 0.04±0.02 | 0.08±0.05 | 0.10±0.07 | 0.12±0.11 | **0.08±0.04** |
| | CT | 0.04±0.03 | 0.05±0.01 | 0.09±0.05 | 0.09±0.06 | 0.11±0.09 | 0.11±0.06 |
| | CRN | 0.04±0.03 | **0.04±0.02** | **0.06±0.03** | 0.08±0.06 | **0.09±0.06** | 0.08±0.03 |
| | RMSN | 0.04±0.02 | 0.05±0.02 | 0.08±0.06 | 0.08±0.04 | 0.11±0.07 | 0.08±0.02 |
| | VCIP | **0.02±0.01** | 0.04±0.03 | 0.07±0.05 | **0.07±0.05** | 0.10±0.08 | 0.10±0.07 |
| $\gamma = 2$ | ACTIN | 0.10±0.02 | 0.15±0.07 | 0.22±0.08 | 0.27±0.12 | 0.32±0.16 | 0.36±0.16 |
| | CT | 0.13±0.05 | 0.21±0.19 | 0.27±0.18 | 0.38±0.30 | 0.42±0.29 | 0.50±0.38 |
| | CRN | 0.10±0.04 | 0.16±0.08 | 0.23±0.10 | 0.28±0.17 | 0.32±0.17 | 0.37±0.23 |
| | RMSN | 0.07±0.02 | 0.11±0.05 | 0.17±0.07 | 0.20±0.09 | 0.31±0.23 | 0.30±0.13 |
| | VCIP | **0.06±0.02** | **0.09±0.06** | **0.13±0.07** | **0.16±0.09** | **0.18±0.13** | **0.19±0.10** |
| $\gamma = 3$ | ACTIN | 0.24±0.07 | 0.40±0.11 | 0.51±0.13 | 0.60±0.11 | 0.66±0.10 | 0.74±0.08 |
| | CT | 0.36±0.12 | 0.54±0.16 | 0.70±0.15 | 0.84±0.17 | 0.97±0.18 | 1.04±0.17 |
| | CRN | 0.23±0.07 | 0.36±0.08 | 0.47±0.10 | 0.57±0.10 | 0.65±0.13 | 0.71±0.12 |
| | RMSN | 0.16±0.05 | 0.25±0.09 | 0.32±0.10 | 0.41±0.14 | 0.51±0.15 | 0.60±0.20 |
| | VCIP | **0.15±0.04** | **0.18±0.06** | **0.29±0.10** | **0.34±0.10** | **0.38±0.09** | **0.38±0.08** |
| $\gamma = 4$ | ACTIN | 0.42±0.14 | 0.71±0.13 | 0.91±0.17 | 1.05±0.21 | 1.19±0.10 | 1.30±0.13 |
| | CT | 0.55±0.20 | 0.88±0.25 | 1.18±0.23 | 1.43±0.36 | 1.58±0.22 | 1.69±0.27 |
| | CRN | 0.38±0.09 | 0.60±0.08 | 0.79±0.13 | 0.92±0.08 | 1.11±0.23 | 1.19±0.18 |
| | RMSN | 0.30±0.10 | 0.45±0.10 | 0.61±0.16 | 0.75±0.16 | 0.89±0.28 | 0.98±0.22 |
| | VCIP | **0.29±0.08** | **0.42±0.13** | **0.53±0.15** | **0.60±0.15** | **0.68±0.19** | **0.75±0.20** |

# E. Appended Results

This section presents additional experimental results of different models on the tumor simulation dataset. The evaluation was conducted from two main aspects focusing on model performance under identical and different intervention strategies.

Under identical intervention strategies (Table 5), the results demonstrate that as the interference level $\gamma$ increases (from 1 to 4), prediction errors of all models show an upward trend, indicating that stronger interference reduces prediction accuracy. The VCIP model achieved optimal performance in most cases, particularly showing significant advantages at higher interference levels ($\gamma$=3,4). As the prediction horizon $\tau$ extends, prediction errors increase across all models, which aligns with the intuition that longer-term predictions are more challenging.

Under different intervention strategies (Table 6), prediction errors increased compared to identical intervention scenarios, reflecting the generalization challenges models face when encountering unseen intervention strategies. VCIP and RMSN models demonstrated better adaptability, especially in short-term predictions ($\tau$=1,2). At high interference levels ($\gamma$=3,4), performance differences between models became more pronounced, with VCIP maintaining relatively stable predictive capabilities.

Figures 7 through 10 show that the subtle consistency between GPR and RCS metrics under both identical and distinct intervention strategies suggests that the ranking process itself may introduce variations in intervention patterns, potentially affecting the comparative assessment of model performance across different evaluation metrics.

# F. Hyperparameter Tuning

The hyperparameter settings used in our experiments are detailed in Table 7. For all baseline models including RMSN, CRN, CT, and ACTIN, we followed the hyperparameter optimization strategy and ranges consistent with those reported in (Wang et al., 2024). Specifically, hyperparameter optimization was conducted through random grid search. Readers can refer to (Wang et al., 2024) for detailed hyperparameter configurations of these baseline models.

*Table 6.* Model comparison results across different interference levels ($\gamma$) and prediction horizons ($\tau$), with distinct intervention strategies applied during training and testing phases. Results are reported as target distance values (mean $\pm$ std over five runs).

| | | $\tau = 1$ | $\tau = 2$ | $\tau = 3$ | $\tau = 4$ | $\tau = 5$ | $\tau = 6$ |
|---|---|---|---|---|---|---|---|
| $\gamma = 1$ | ACTIN | 0.15$\pm$0.09 | 0.18$\pm$0.10 | 0.27$\pm$0.20 | 0.33$\pm$0.22 | 0.40$\pm$0.28 | 0.45$\pm$0.33 |
| | CT | 0.17$\pm$0.07 | 0.23$\pm$0.10 | 0.38$\pm$0.12 | 0.45$\pm$0.18 | 0.50$\pm$0.17 | 0.59$\pm$0.23 |
| | CRN | 0.12$\pm$0.04 | 0.15$\pm$0.04 | **0.19$\pm$0.07** | 0.27$\pm$0.08 | 0.30$\pm$0.07 | 0.36$\pm$0.09 |
| | RMSN | 0.10$\pm$0.03 | **0.14$\pm$0.02** | 0.20$\pm$0.05 | 0.28$\pm$0.10 | 0.33$\pm$0.12 | 0.36$\pm$0.09 |
| | VCIP | **0.10$\pm$0.05** | **0.14$\pm$0.07** | **0.19$\pm$0.10** | **0.27$\pm$0.15** | **0.24$\pm$0.11** | **0.22$\pm$0.08** |
| $\gamma = 2$ | ACTIN | 0.17$\pm$0.05 | 0.28$\pm$0.07 | 0.41$\pm$0.08 | 0.47$\pm$0.09 | 0.60$\pm$0.18 | 0.60$\pm$0.16 |
| | CT | 0.21$\pm$0.08 | 0.32$\pm$0.11 | 0.49$\pm$0.15 | 0.58$\pm$0.17 | 0.70$\pm$0.26 | 0.74$\pm$0.28 |
| | CRN | 0.19$\pm$0.05 | 0.30$\pm$0.11 | 0.41$\pm$0.16 | 0.49$\pm$0.19 | 0.62$\pm$0.28 | 0.60$\pm$0.21 |
| | RMSN | 0.13$\pm$0.06 | **0.20$\pm$0.08** | 0.29$\pm$0.07 | 0.39$\pm$0.12 | **0.53$\pm$0.23** | 0.50$\pm$0.17 |
| | VCIP | **0.11$\pm$0.06** | 0.25$\pm$0.15 | **0.25$\pm$0.12** | **0.38$\pm$0.30** | 0.55$\pm$0.36 | **0.35$\pm$0.10** |
| $\gamma = 3$ | ACTIN | 0.44$\pm$0.35 | 0.67$\pm$0.45 | 0.90$\pm$0.64 | 0.96$\pm$0.58 | 0.98$\pm$0.56 | 1.10$\pm$0.58 |
| | CT | 0.52$\pm$0.13 | 0.81$\pm$0.21 | 1.11$\pm$0.38 | 1.27$\pm$0.47 | 1.32$\pm$0.36 | 1.41$\pm$0.39 |
| | CRN | 0.31$\pm$0.09 | 0.53$\pm$0.14 | 0.71$\pm$0.22 | 0.82$\pm$0.22 | 0.91$\pm$0.28 | 1.04$\pm$0.33 |
| | RMSN | 0.22$\pm$0.06 | **0.35$\pm$0.10** | 0.50$\pm$0.18 | 0.55$\pm$0.10 | 0.59$\pm$0.06 | 0.69$\pm$0.09 |
| | VCIP | **0.17$\pm$0.04** | 0.39$\pm$0.24 | **0.42$\pm$0.25** | **0.49$\pm$0.27** | **0.49$\pm$0.16** | **0.62$\pm$0.25** |
| $\gamma = 4$ | ACTIN | 0.45$\pm$0.13 | 0.78$\pm$0.09 | 1.00$\pm$0.11 | 1.15$\pm$0.08 | 1.38$\pm$0.10 | 1.49$\pm$0.07 |
| | CT | 0.62$\pm$0.19 | 0.93$\pm$0.21 | 1.30$\pm$0.27 | 1.56$\pm$0.25 | 1.81$\pm$0.30 | 1.98$\pm$0.28 |
| | CRN | 0.47$\pm$0.11 | 0.69$\pm$0.10 | 0.90$\pm$0.13 | 1.03$\pm$0.16 | 1.25$\pm$0.13 | 1.30$\pm$0.20 |
| | RMSN | 0.37$\pm$0.11 | 0.53$\pm$0.07 | 0.68$\pm$0.11 | 0.81$\pm$0.14 | 0.96$\pm$0.22 | 1.04$\pm$0.24 |
| | VCIP | **0.29$\pm$0.04** | **0.44$\pm$0.06** | **0.61$\pm$0.11** | **0.71$\pm$0.09** | **0.77$\pm$0.11** | **0.87$\pm$0.10** |

*Table 7.* Specified ranges for hyperparameter tuning of VCIP across various datasets.

| Hyperparameter | Range (tumor) | Range (MIMIC-III) |
|---|---|---|
| Learning rate $l$ | 0.01, 0.001 | 0.01, 0.001 |
| Minibatch size | 128, 256, 512, 1024 | 128, 256, 512, 1024 |
| Representation size | 8, 12, 16 | 8, 16, 32 |
| hidden size (Generative model) | 8, 12, 16 | 8, 16, 32 |
| FC hidden units (Generative model) | 8, 12, 16 | 8, 16, 32 |
| hidden size (Inference model) | 8, 12, 16 | 8, 16, 32 |
| FC hidden units (Inference model) | 8, 12, 16 | 8, 16, 32 |
| Dropout rate | 0, 0.1, 0.2, 0.3 | 0, 0.1, 0.2, 0.3 |
| Random search iterations | 30 | 30 |
| Number of epochs | 100 | 100 |

# G. Limitations

Our approach hinges on the standard causal identification assumptions of consistency, positivity and sequential ignorability. In particular, positivity requires that each candidate intervention has sufficient support in the observational data, and violations of this assumption can lead to severe degradation in optimization performance. Sequential ignorability assumes that all confounders affecting both treatment assignment and outcomes are observed and correctly modeled; unobserved confounding remains an open challenge and may introduce bias into both estimation and downstream planning.

Beyond these identification requirements, practical deployment poses additional challenges. The variational inference and sequential planning stages are computationally intensive, making real-time decision support and scaling to large cohorts or long horizons difficult without further algorithmic improvements. Clinical time-series data are often sparse, irregularly sampled and plagued by missing values; while imputation and generative augmentation can help, they introduce extra hyperparameters and potential sources of bias. Finally, personalized intervention policies must be accompanied by rigorous fairness, transparency and privacy safeguards—ethical and governance issues that our current framework does not explicitly address but which are critical for real-world applications.

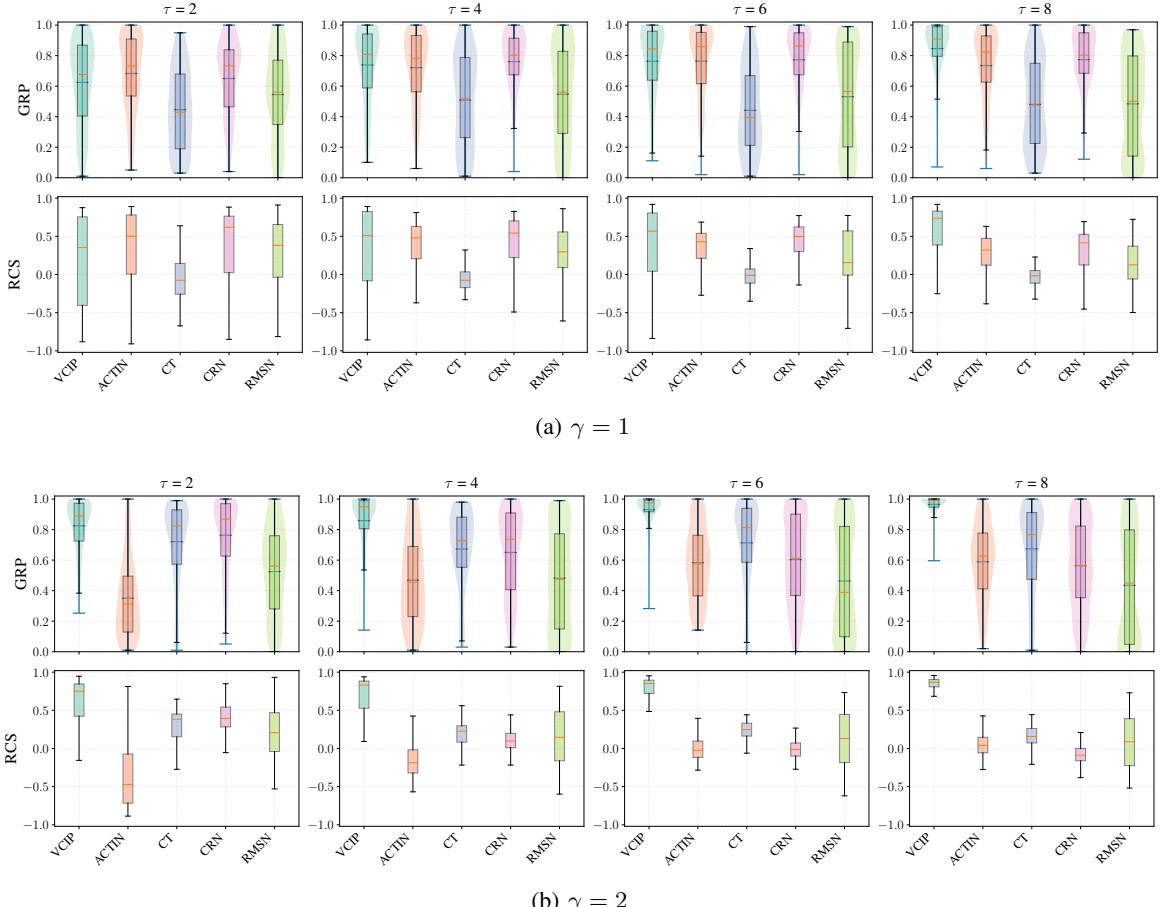

*Figure 7.* Evaluation results (Part 1) on the tumor simulation dataset with varying prediction horizons ($\tau = 2, 4, 6, 8$) under different confounding bias levels ($\gamma = 1, 2$). The real intervention sequence in test set follows the identical intervention strategy as in training set. The figures show GRP metrics that measure model performance, where higher values indicate better predictive capability.

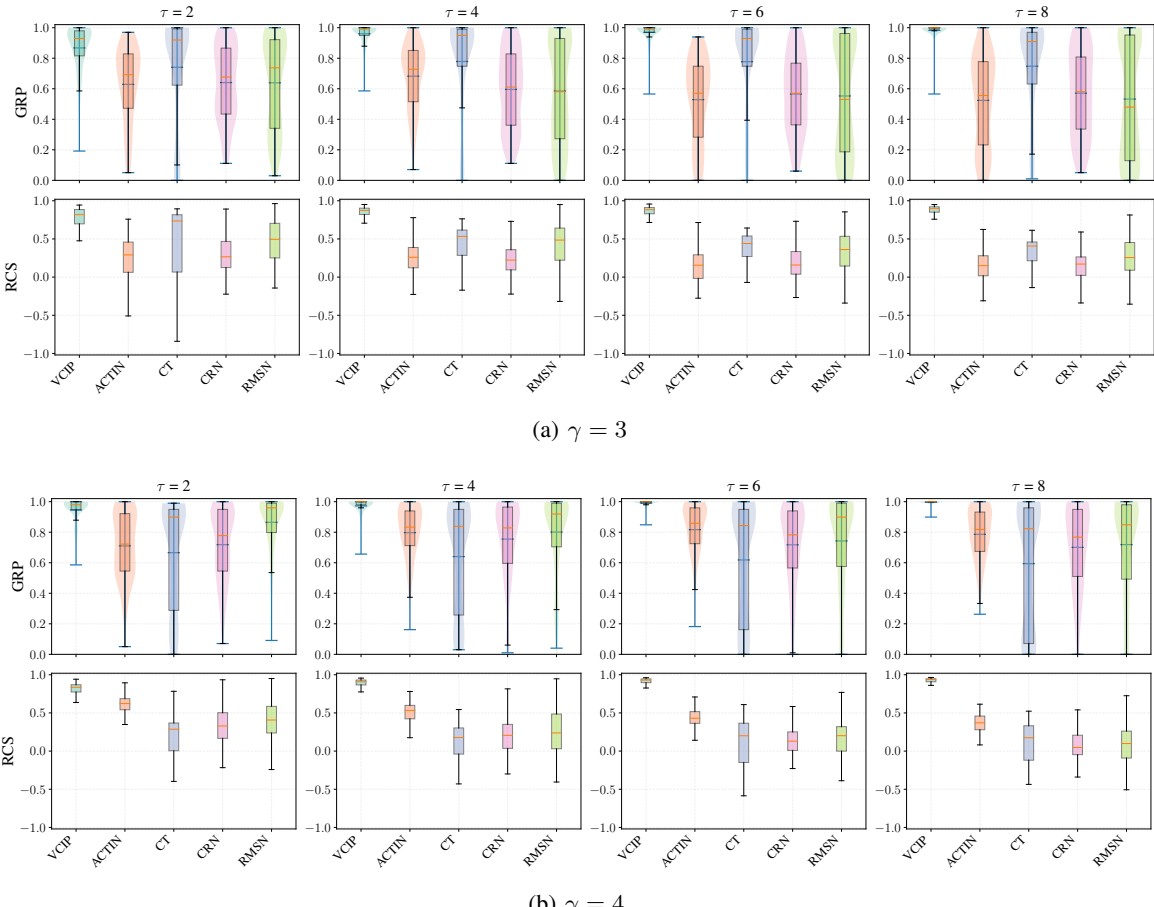

*Figure 8.* Evaluation results (Part 2) on the tumor simulation dataset with varying prediction horizons ($\tau = 2, 4, 6, 8$) under different confounding bias levels ($\gamma = 3, 4$).

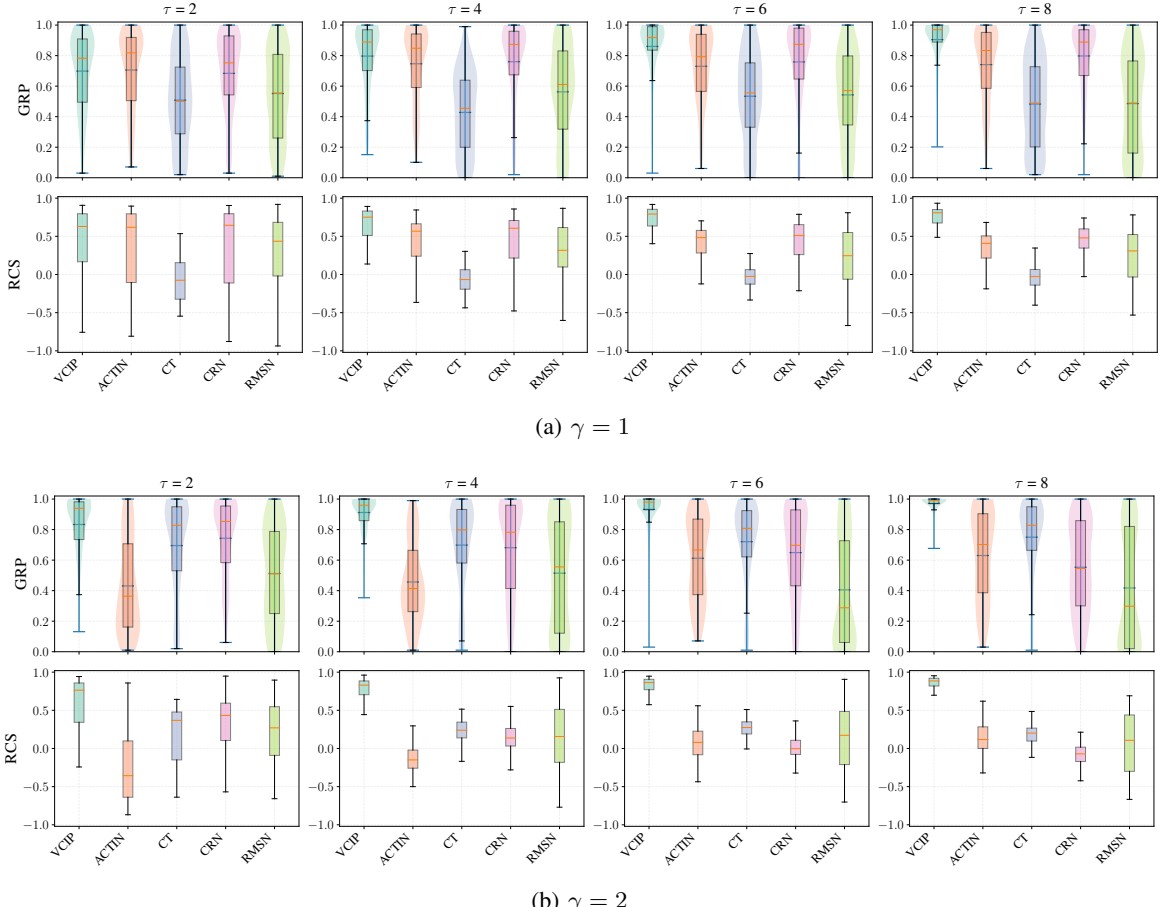

*Figure 9.* Evaluation results (Part 1) on the tumor simulation dataset with varying prediction horizons ($\tau = 2, 4, 6, 8$) under different confounding bias levels ($\gamma = 1, 2$). The real intervention sequence in test set follows a different intervention strategy from training set. The figures show GRP metrics that measure model performance, where higher values indicate better predictive capability.

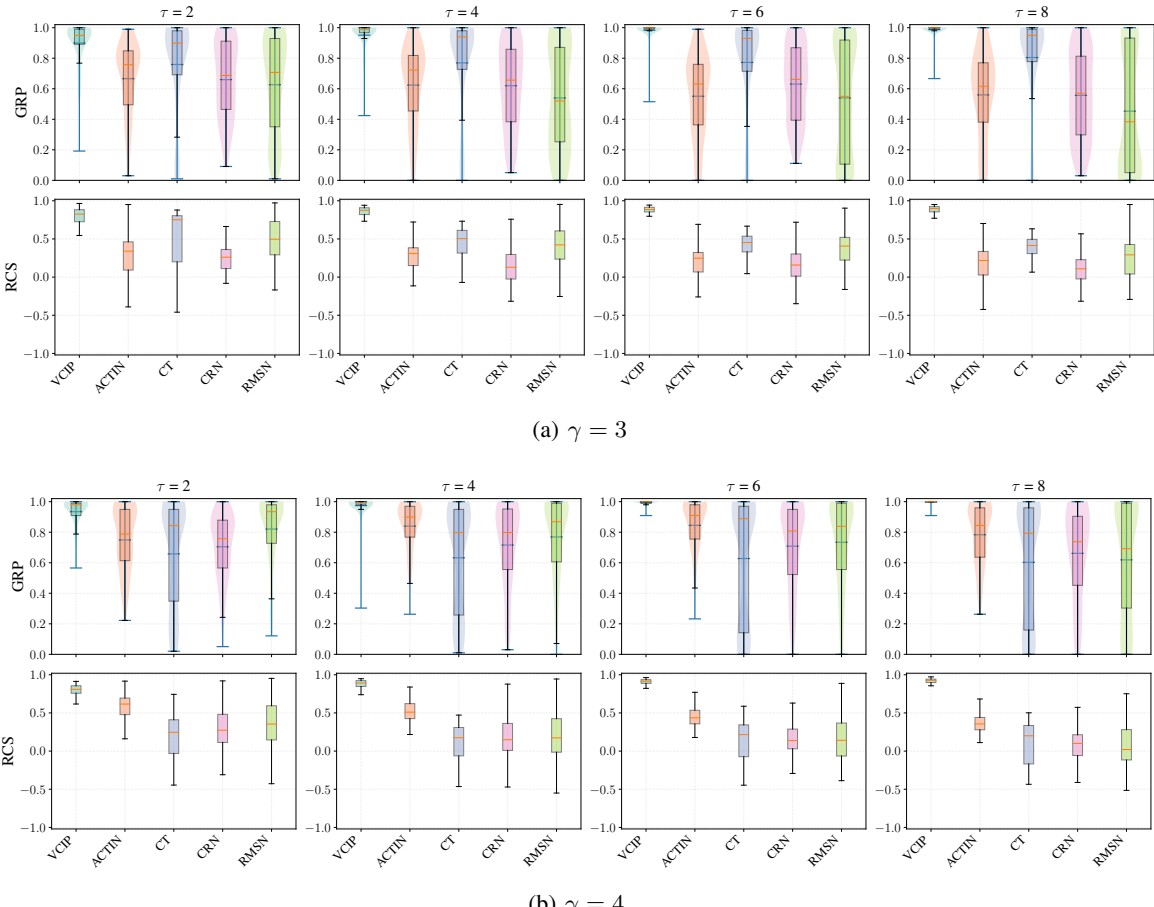

*Figure 10.* Evaluation results (Part 2) on the tumor simulation dataset with varying prediction horizons ($\tau = 2, 4, 6, 8$) under different confounding bias levels ($\gamma = 3, 4$).

