# OpenReview forum: "Variational Counterfactual Intervention Planning to Achieve Target Outcomes"
_ICML.cc/2025/Conference — ICML 2025 poster_

### Official Review · Reviewer_ixW5 · 2025-03-03

**Overall Recommendation:** 3

**Summary:**

The  paper introduces Variational Counterfactual Intervention Planning (VCIP), a framework for determining optimal intervention sequences in personalized healthcare and other temporal decision-making systems.

**Claims And Evidence:**

yes

**Essential References Not Discussed:**

no

**Experimental Designs Or Analyses:**

seem reasonable

**Methods And Evaluation Criteria:**

they do

**Other Comments Or Suggestions:**

see below

**Other Strengths And Weaknesses:**

Pros :

- Addresses a critical problem in personalized healthcare—predicting the best intervention sequences rather than merely estimating outcomes.
- Reduces compounding errors common in standard counterfactual estimations by directly modeling target achievement probability.
- Outperforms baseline models in both simulated and real-world datasets, particularly in ranking interventions.
- Uses principled causal inference via the g-formula and variational inference, ensuring theoretically sound predictions.
- Avoids over-reliance on counterfactual predictions, which are inherently unobservable and prone to error accumulation.

Cons:
- Dependence on quality of observational data—errors in training data can propagate through the model.
- In medicine usually there is no single target we want to optimise for, but rather a range of values, as such the existence or validity of the target outcome is in question
- Handling high-dimensional intervention spaces or multiple simultaneous treatments may be computationally expensive.

**Questions For Authors:**

How does VCIP handle unseen interventions? Since it relies on observational data, does it generalize well when encountering interventions not present in training data?

How scalable is VCIP to high-dimensional, multi-treatment scenarios? The paper focuses on limited intervention sequences—can this approach scale efficiently to complex, real-world medical decisions?

**Relation To Broader Scientific Literature:**

builds upon literature for causal reasoning in personalised healthcare

**Theoretical Claims:**

checked but not proofs in appendix

---

> ### Author Rebuttal · Authors · 2025-03-31
>
> Thank you for your thorough review and recognition of this work. Below we address your main concerns:
>
> **Regarding Concerns about Data:**
>
> "Dependence on observational data quality" is a common challenge in causal inference, especially in medical data analysis where EHR data usage can introduce quality issues. The VCIP framework mitigates error propagation by using variational inference to introduce latent variables that capture system state evolution, and directly modeling target achievement probability rather than relying on explicit predictions. We utilize the g-formula to establish connections between intervention and observational distributions, enabling reliable training on observational data. Nevertheless, we acknowledge the importance of data preprocessing and will explore data cleaning and specific medical data processing strategies in future work to enhance model robustness when facing imperfect data.
>
> **Regarding Concerns about Prediction Intervals:**
>
> Thank you for this valuable comment. In practice, many clinical decisions indeed aim to maintain patient metrics within "safe ranges." We acknowledge that our current model's focus on single target point optimization is a limitation, but a straightforward approach would be taking a value from the "range of values" as the target, such as the midpoint of the interval. For more precise characterization, extending VCIP to model the probability of falling within a target range would be necessary. While this is beyond the current paper's scope, it represents an interesting research direction that could effectively enhance our method's clinical applicability.
>
> **Regarding Concerns about High-Dimensionality:**
>
> This paper validates using the widely-used real-world MIMIC-III dataset, which includes 25 time-varying features and a 2-dimensional intervention space, representing a relatively complex dataset in current medical causal inference research. At this scale, VCIP's complete runtime (including training and intervention ranking) is approximately 2700 seconds, which falls within an acceptable range. Validating VCIP's scalability in higher-dimensional intervention spaces requires newer, more complex datasets. We acknowledge this has important application value, but building such datasets in the medical field faces significant challenges. Future work will explore optimizing algorithm efficiency and testing performance in more complex medical decision scenarios to further validate the method's practicality.
>
> **Regarding Concerns about Unseen Interventions:**
>
> In fact, to ensure identifiability of causal effects, this paper makes standard assumptions, including Assumption 2 (Sequential Overlap), which theoretically guarantees that any intervention has a possibility of being observed (note that "unseen" in this paper refers to using different intervention strategies in testing than in training, which doesn't conflict with Assumption 2). This assumption enables good generalization when observational data is sufficient. When this assumption is violated, meaning interventions not present in training data are encountered, we conducted experiments on the Tumor dataset. Specifically, with $\gamma=4$, we set the probability of interventions greater than 0.5 to 0 (this may be simple but effectively violates Assumption 2). Below are the Optimization experiment results:
>
> | Method           | τ=2           | τ=4           | τ=6           |
> | ---------------- | ------------- | ------------- | ------------- |
> | RMSN (violated)  | $2.24\pm0.90$ | $3.41\pm1.26$ | $4.49\pm1.17$ |
> | RMSN (satisfied) | $0.45\pm0.10$ | $0.75\pm0.16$ | $0.98\pm0.22$ |
> | VCIP (violated)  | $1.32\pm0.30$ | $2.10\pm0.40$ | $2.81\pm0.46$ |
> | VCIP (satisfied) | $0.42\pm0.13$ | $0.60\pm0.15$ | $0.75\pm0.20$ |
>
> When Assumption 2 is violated, both models face unseen intervention sequences. As shown, both VCIP and RMSN performance deteriorates, but VCIP exhibits a smaller performance drop, indicating superior generalization capabilities.

---

### Official Review · Reviewer_g91d · 2025-03-10

**Overall Recommendation:** 3

**Summary:**

This paper addresses the problem of time varying treatment effect, aiming at finding the sequence of treatments that optimize a target outcome, instead of the typical problem of predicting potential outcomes. It uses the g-formula and a variational approach to estimate the conditional likelihood of achieving target outcomes. This is then used to find an optimal sequence of treatments.

## Update after rebuttal
After the substantive discussion with the authors, who tried to clarify my main points of contention, and in light of the interesting and important topic, with promising experimental results, I am willing to change my score from 2: Weak Reject, to 3: Weak Accept.

I would like to thank the authors for their time and effort in this discussion.

**Claims And Evidence:**

Proofs are offered for the main claims made in the paper. Some of them are, in my opinion, problematic. I deepen in that in the section of Theoretical Claims.

**Essential References Not Discussed:**

All important papers are cited.

**Experimental Designs Or Analyses:**

Overall, experiments look sound, but I didn’t check the code.

It would have been interesting to see how the proposed model compares to other models when estimating counterfactual outcomes, with distance metrics to ground truth counterfactuals. The estimated means of the output could be used to do that, and it would probably give a better idea of how trustable the model is.

**Methods And Evaluation Criteria:**

The idea of the rankings is consistent with the main objective of the paper. However, more direct comparisons with other benchmarks in traditional metrics like distance between counterfactual outcomes and ground truth counterfactuals would probably strengthen the paper.

Apart from this, it is a bit difficult to understand the content of the tables from the captions.

**Other Comments Or Suggestions:**

-

**Other Strengths And Weaknesses:**

The paper addresses an important problem, and the idea seems interesting. However, there are some important issues that need to be clarified. Also, the paper is difficult to understand, and the authors could have done more efforts to improve clarity and give more explanations, for example on theorem 4.1 and the appendix proofs, which are fundamental for the paper despite being in the appendix. Also, it would be interesting if some intuitive interpretation of the terms in eq.6 could be given.

**Questions For Authors:**

The treatment sequence is optimized with gradient descent. Does the method offer a solution for categorical treatments? Is it explained in the paper?

**Relation To Broader Scientific Literature:**

In general, previous important works like Causal Transformer, Counterfactual Recurrent Networks or ACTIN are properly discussed.

**Theoretical Claims:**

I Checked the theoretical claims and proofs, and I have several concerns:

c1) I am not convinced by theorem 4.1, which is fundamental in this work. While the relations expressed in the equations are correct, I am not sure that optimizing the expression of eq. 6 amounts to optimizing eq. 2, which is the main claim. The problem that I see is that, if ELBO1 is not maximized, then \epsilon_{1} can be arbitrarily large. Then, the error between the interventional loglikelihood (eq. 2) and the expression in eq. 6 will also be arbitrarily large. It would be very good if the authors could address this problem, as this is a major concern.

c2) In 4.1, it is mentioned that a variational distribution is introduced to approximate the true posterior, but the term do(a) is omitted for practical considerations, as its effects are partially captured by Y. However, in light of the kind of problem that the paper addresses this seems that it is an important approximation that is not very discussed. Maybe the authors could try to better justify this approximation.

c3) In the inference model, the claim that Z_{s} is obtained from its descendants does not seem very convincing. As mentioned in the last lines of this section, there is variable Z’_{s} (one from each step) obtained from the Z’_{s-1} and a_{s-1}. Then, those are the variables, for s={t+1,….t+\thau}, used to obtain Z_{s}. Then, I think it would be better to say that Z_{s} is obtained from the descendant treatments after time s than from the descendants of latent factors.

---

> ### Author Rebuttal · Authors · 2025-03-31
>
> Thank you for your thoughtful review. Below we address your main concerns:
>
> **Regarding Concerns about Evaluation Criteria:**
>
> Our primary contribution is a novel "counterfactual target achievement" problem formulation, which differs fundamentally from counterfactual estimation addressed by models like RMSN. This difference in objectives explains why we didn't initially compare models on counterfactual estimation tasks.
>
> Following your valuable suggestion, we conducted additional multi-step estimation experiments at γ=2,4 to better illustrate how VCIP compares with other models on counterfactual prediction:
>
> |           | τ=4       | τ=6     | τ=8     | τ=10    |
> | --------- | --------- | --------- | --------- | --------- |
> | RMSN  γ=2 | 0.90±0.23 | 1.07±0.32 | 1.18±0.36 | 1.37±0.49 |
> | VCIP  γ=2    | 0.90±0.70 | 0.84±0.67 | 0.84±0.62 | 0.79±0.65 |
> | RMSN γ=4  | 1.34±0.21 | 1.61±0.31 | 1.85±0.32 | 2.09±0.43 |
> | VCIP  γ=2   | 1.83±0.61 | 1.75±0.66 | 1.74±0.59 | 1.79±0.64 |
>
> As Figure 1 shows, using counterfactual predictions is suboptimal for counterfactual target achievement, as treating $Y\_{target}$ as an intermediate variable causes compounding errors. Our method avoids this by directly incorporating $Y\_{target}$ into the likelihood. The experiments reveal interesting insights:
>
> - Even when VCIP's counterfactual estimations sometimes underperform RMSN, it still shows advantages in the achievement problem, validating our framework's effectiveness.
>
> - Accurate counterfactual estimation helps the achievement problem: at τ=2, where RMSN outperforms VCIP in estimation, the performance gap in our problem is small (approximately 0.02), while at τ=12, where VCIP excels in estimation, the achievement performance gap widens (approximately 0.4).
>
> **Regarding concerns about Theoretical Claims**:
>
> C1)
>
> While $\mathrm{ELBO}\_1$ captures the true causal mechanism through consideration of interventions (do-operator), in practical training we typically only have observational data for maximum likelihood estimation. As long as the model structure has sufficient expressivity and the observational data reasonably approximates the causal process, $\mathrm{ELBO}\_1$ can be indirectly approximated during learning, making $\epsilon\_1$ relatively small. This is supported by our ablation studies (Table 3): even without adjustment, VCIP performs comparably to or better than RMSN, indicating that maximizing eq. 5 (where $\mathrm{ELBO}\_2$ can be directly optimized based on observational data) effectively drives improvements in eq. 2 ($\mathrm{ELBO}\_1$), thus approximating $\mathcal{O}$ and preventing $\epsilon\_1$ from becoming "arbitrarily large."
>
> For intuitive interpretation of eq. 6, term A maximizes observational likelihood, while terms B and C serve as adjustment terms:
>
> $$
> \text{(A)}\ \mathrm{ELBO}\_2\ \ +\ \text{(B)}-\ \sum\_{s=t}^{t+\tau-1}\mathbb E\_{q\_\phi}[\log p\_\theta(\mathbf{a}\_s\mid \mathbf{Z}\_s)]\ +\ \text{(C)}\ \log p\_\theta(\bar{\mathbf{a}}\_{t,\tau}\mid \bar{\mathbf{H}}\_t),
> $$
>
> Increasing term B encourages the model to learn states that cannot accurately predict interventions, intuitively allowing interventions to "break free" from observational distribution relationships, thus mitigating confounding bias. Term C compensates for inherently reasonable action sequences by providing a bonus, avoiding excessive penalties in term B.
>
> This demonstrates our core intuition of jointly optimizing these three terms to elegantly optimize $\mathrm{ELBO}\_1$ through $\mathrm{ELBO}\_2$ combined with appropriate penalties and bonuses for action sequences.
>
> To verify the effects of B and C separately, we conducted more detailed ablation experiments.
>
> |        | GRP τ=2 | RCS τ=4 | Optimization τ=6 | τ=8  | τ=10 | τ=12 |
> | ------ | ------- | ------- | ---------------- | ---- | ---- | ---- |
> | Ours   | 0.94    | 0.87    | 0.75             | 0.92 | 0.97 | 1.08 |
> | w/o B  | 0.92    | 0.83    | 0.77             | 0.95 | 1.00 | 1.18 |
> | w/o C  | 0.91    | 0.61    | 0.74             | 0.91 | 0.92 | 1.01 |
> | w/o BC | 0.76    | 0.60    | 0.91             | 1.14 | 1.27 | 1.48 |
>
> From the results, we can see that both B and C improve performance on Ranking and Optimization tasks. However, considering both tasks comprehensively, incorporating both B and C simultaneously is the most appropriate approach.
>
> C2)
>
> Please refer to our response to reviewer XzCZ's "Regarding Concerns about Claims And Evidence" section.
>
> C3)
>
> We will revise our description of distribution $q_\phi$ in the updated manuscript.
>
> **Regarding Concerns about Categorical Interventions**
>
> We didn't design additional optimization algorithms for categorical treatments. Since categorical treatments are typically enumerable, one can use the ranking approach and enumerate to find the optimal intervention sequence. For more complex categories, techniques like Gumbel-Softmax could be used to design optimization processes similar to Algorithm 1.

---

> > ### Comment · Reviewer_g91d · 2025-04-03
> >
> > I would like to thank the authors for addressing my main concerns, and performing additional experiments following my suggestions.
> >
> > While the “counterfactual target achievement” is an interesting problem formulation, I am still dubious about the proposed variational approach and the claim that it offers theoretical guarantees.
> >
> > In the rebuttal, the authors say that ELBO1 can be indirectly approximated if the observational data reasonably approximates the causal process; however, I think that the proposed variational approach does not have proper causal guarantees, unlike other methods such as RMSN, CRN or G-Net, that can estimate potential outcomes if the assumptions of consistency, overlap and ignorability are fulfilled. As you mention, the proposed approach has the additional restriction that the observed data must reasonably approximate the causal process.
> >
> > Although a model can show interesting results despite not having a proper causal adjusting, I think that this lack of theoretical guarantees should be clearly mentioned and discussed. On the other hand, to my understanding, the method for finding optimal treatment sequences consists of optimizing an ELBO (which seems more observational than causally adjusted) depending on treatments through gradient descent. I think that this same optimization process could have been applied to optimize exact likelihood measures of other methods with more guarantees, and I am still not sure of the advantages of using the authors’ variational approach over these other methods.

---

> > > ### Author Response · Authors · 2025-04-03
> > >
> > > We appreciate the reviewer's thoughtful feedback and the opportunity to clarify these important points.
> > >
> > > **First**, we want to emphasize that VCIP is indeed built upon the standard causal assumptions of consistency, ignorability, and overlap, as stated in Appendix A of our paper. The requirement that "observed data must reasonably approximate the causal process" is fundamentally consistent with these common assumptions: when there are unobserved confounders or severe non-overlap issues, any causal method based on observational data—including RMSN, CRN, and G-Net—would struggle to effectively learn intervention effects. This is confirmed by our experimental results on positivity violations discussed in our response to reviewer XzCZ under "Regarding Concerns about Claims and Evidence."
> > >
> > > In fact, Table 8 in the CRN paper demonstrates that even without adversarial balancing representation strategies, the model can still achieve reasonable predictive performance on counterfactual estimation. This aligns with what we observe in the VCIP framework: although VCIP's derivation is based on the do-operator, in practice, training with just the observational distribution can indirectly optimize the model. The fundamental reason is that as long as the observational and interventional distributions are not completely disconnected, the model can learn information relevant to the true intervention effects from observational data. In other words, estimations without additional balancing or weighting are not valueless; the natural diversity of interventions and states in observational data itself provides meaningful information for the model. Of course, incorporating additional balancing or adversarial corrections can further reduce estimation bias, but even without such "adjustments," models typically capture some of the true intervention mechanisms. VCIP leverages this "overlap between observational and interventional distributions" to indirectly approximate ELBO₁ while maximizing observational likelihood through moderate interventional distribution adjustments (e.g., the regulatory term in Eq. 6), thereby achieving effective results in target achievement tasks.
> > >
> > > **Second**, we apologize for not clearly explaining how other counterfactual estimation models optimize interventions. We also use gradient descent to optimize intervention sequences as in Algorithm 1, but with the objective of minimizing expected loss (as can be seen in our code at `src/baselines/time_varying_model/optimize_interventions_onetime`, optimization details will be added to the revised paper):
> > > $$
> > > \min\_{\bar{\mathbf{a}}\_{t,\tau}}\ \Bigl\|\hat{\mathbf{Y}}[\bar{\mathbf{a}}\_{t,\tau}] - \mathbf{Y}\_{\mathrm{target}}\Bigr\|
> > > $$
> > >
> > > However, as demonstrated in our case studies (e.g., Figure 5) in the paper and Figure 1, due to the cumulative nature of prediction errors $\|\hat{\mathbf{Y}} - \mathbf{Y}\|$, this metric $\|\hat{\mathbf{Y}}[\bar{\mathbf{a}}\_{t,\tau}] - \mathbf{Y}\_{\mathrm{target}}\|$ cannot guarantee synchronization with the true $\|\mathbf{Y}[\bar{\mathbf{a}}\_{t,\tau}] - \mathbf{Y}\_{\mathrm{target}}\|$. The two may diverge in critical regions. This means that methods relying on "first predicting potential outcomes, then comparing target distances" face increased optimization challenges—when prediction errors cannot be promptly corrected, they lead to deviations between selected interventions and the truly optimal strategy.
> > >
> > > In contrast, VCIP directly incorporates $\mathbf{Y}\_{\mathrm{target}}$ into the likelihood (ELBO) during training. This means the model no longer needs to "go around" by first predicting the final outcome, but instead "directly" evaluates the possibility of target achievement. VCIP strengthens the feedback on "whether the final outcome can approach $\mathbf{Y}\_{\mathrm{target}}$" during training without explicit regression on potential outcomes. This approach suppresses the accumulation of intermediate prediction errors and tightly couples target achievement with the model's optimization objective, resulting in better performance.
> > >
> > > We hope these clarifications address your concerns, and we sincerely appreciate your valuable time and expertise in reviewing our work.

---

### Official Review · Reviewer_XzCZ · 2025-03-11

**Overall Recommendation:** 3

**Summary:**

The paper introduces an approach named "variational counterfactual intervention planning (VCIP)" to address the problem of optimal sequences of interventions selection towards a target outcome. The method is useful particularly in healthcare scenarios. Traditional counterfactual estimation methods suffer from compounding errors due to their inherent reliance on unobservable counterfactual outcomes. VCIP addresses this issue by reformulating the problem through a variational inference framework, directly modeling the conditional likelihood of achieving target outcomes, hence avoiding explicit prediction of counterfactuals. Experiments are conducted on both synthetic and real-world datasets, demonstrated superior performance compared to existing methods.

**Claims And Evidence:**

The following claims need further evidence or clarity to fully support the assertions in a convincing way:

1. Robustness under violations of standard causal assumptions
The paper implicitly assumes consistency, positivity, and sequential ignorability without extensive empirical or theoretical exploration of sensitivity to these assumptions. Real-world scenarios often violate these assumptions due to unobserved confounding, missing data, measurement error, or non-random treatment assignment patterns. The paper should explicitly evaluate or at least discuss VCIP’s performance under potential assumption violations. (Perhaps, sensitivity analysis, or ablation studies?)
2. Practical applicability in personalized healthcare scenarios
Authors claim significant potential impact in personalized healthcare but provide limited discussion of realistic challenges such as computational complexity, data sparsity, missingness, and ethical issues.

**Essential References Not Discussed:**

No.

**Experimental Designs Or Analyses:**

The experimental design is methodologically robust, clear, and aligned with standard practices. However, I identified some issues:

The experimental design implicitly relies on approximations for intractable intervention sequences. While reasonable, this decision is not empirically validated, raising minor suspicion about possible biases or implications of this simplification.

Also, the paper lacks of robustness analysis against violations of standard assumptions.

**Methods And Evaluation Criteria:**

Yes

**Other Comments Or Suggestions:**

None

**Other Strengths And Weaknesses:**

None

**Questions For Authors:**

None

**Relation To Broader Scientific Literature:**

The paper’s key contributions relate closely to several well-established strands within the broader scientific literature on causal inference and variational inference.  In contrast, this paper introduces a novel inverse formulation—the "counterfactual target achievement" problem—shifting the goal from prediction to actively selecting intervention sequences that drive outcomes toward specified targets.

**Theoretical Claims:**

I carefully checked the statements of theoretical claims provided in the paper but did not go into the detailed verification of the proofs.

In line 145, the authors explicitly mention that they "omit the intractable intervention sequence $do(\bar{a}_{t,\tau})$ as its effects are partially captured in $Y_{t+\tau}$. This step, while practically understandable, might slightly weaken the theoretical rigor. A suspicious aspect here is whether this approximation significantly affects the theoretical guarantees or introduces unacknowledged biases. This step is not fully justified, which leaves room for questioning the precision or generality of theoretical guarantees.

In Theorem 4.1, why can we safely assume $\epsilon_1$ and $\epsilon_2$ are positive? Any justification?

---

> ### Author Rebuttal · Authors · 2025-03-31
>
> Thank you for the thoughtful review and for acknowledging our work. We will address your main concerns below.
>
> **Regarding Concerns about Claims and Evidence:**
>
> The **Consistency** assumption is typically satisfied in clinical settings where treatments are well-defined and outcomes can be stably measured after treatment administration.
>
> To test model performance under Positivity assumption violations, we conducted experiments on the Tumor dataset at $\gamma$=4, setting the probability of receiving treatment to 0 when treatment values >0.5 (while simple, this effectively violates the Positivity assumption). Below are results under the Optimization experiment :
>
> | Method           | τ=2           | τ=4           | τ=6           |
> | ---------------- | ------------- | ------------- | ------------- |
> | RMSN (violated)  | $2.24\pm0.90$ | $3.41\pm1.26$ | $4.49\pm1.17$ |
> | RMSN (satisfied) | $0.45\pm0.10$ | $0.75\pm0.16$ | $0.98\pm0.22$ |
> | VCIP (violated)  | $1.32\pm0.30$ | $2.10\pm0.40$ | $2.81\pm0.46$ |
> | VCIP (satisfied) | $0.42\pm0.13$ | $0.60\pm0.15$ | $0.75\pm0.20$ |
>
> As shown, violating the Positivity assumption degrades performance for both RMSN and VCIP, as models encounter interventions with no supporting observational data during optimization. However, VCIP's performance degrades notably less than RMSN's, likely because RMSN suffers more severely from compounding errors in such scenarios.
>
> Testing robustness to **Sequential Ignorability** violations requires specialized datasets with unobserved confounding, which we couldn't explore due to time constraints. Such violations introduce bias, and while approaches like Time Series Deconfounder exist, resolving this in our context remains an open question outside our current scope. We will gradually explore these directions in our subsequent research work.
>
> Additionally, we acknowledge personalized healthcare applications face multifaceted challenges. Computationally, VCIP requires 2400s for running, and RMSN 4200s, demonstrating clinical feasibility while scaling remains challenging. Data sparsity issues can be addressed through multiple imputation, GAN augmentation, and transfer learning. We recognize the importance of ethical considerations including fairness, transparency, and privacy protection. The revised version will elaborate on these challenges, analyzing computational efficiency, methods for handling sparse data, and expanding ethical discussions to enhance practical applicability.
>
> **Regarding Concerns about Theoretical Claims:**
>
> When we **omit** $do(\bar{\mathbf{a}}\_{t,\tau})$, the variational distribution does not explicitly model the intervention process, but if $\mathbf{Y}\_{t+\tau}$ is of high quality (in experimental settings, sufficient and accurate $\mathbf{Y}\_{t+\tau}$ information can often be observed), the impact brought by the intervention is largely "reflected" in $\mathbf{Y}\_{t+\tau}$. To illustrate this point, we compare by explicitly adding observed intervention information, namely using $q\_{\phi}(\bar{\mathbf{Z}}\_{t,\tau+1} \mid \bar{\mathbf{H}}\_t, \mathbf{Y}\_{t+\tau},\bar{\mathbf{a}}\_{t,\tau})$
>
> 1. **Optimization under $\gamma=3$**
>
>    |                                      | $\tau=2$      | $\tau=6$      | $\tau=8$      | $\tau=12$     |
>    | ------------------------------------ | ------------- | ------------- | ------------- | ------------- |
>    | omit $do(\bar{\mathbf{a}}\_{t,\tau})$ | $0.39\pm0.26$ | $0.63\pm0.33$ | $0.67\pm0.33$ | $0.79\pm0.33$ |
>    | with $\bar{\mathbf{a}}\_{t,\tau}$     | $0.37\pm0.23$ | $0.60\pm0.25$ | $0.64\pm0.23$ | $0.78\pm0.28$ |
>
>    As can be seen, compared to omitting $do(\bar{\mathbf{a}}\_{t,\tau})$, explicitly introducing $\bar{\mathbf{a}}\_{t,\tau}$ shows a minor improvement in model performance, but the difference is not significant.
>
> 2. **Ranking under $\gamma=4$**
>
>    |                                      | GRP $\tau=2$  | RCS $\tau=2$  |
>    | ------------------------------------ | ------------- | ------------- |
>    | omit $do(\bar{\mathbf{a}}\_{t,\tau})$ | $0.94\pm0.09$ | $0.77\pm0.21$ |
>    | with $\bar{\mathbf{a}}\_{t,\tau}$     | $0.95\pm0.09$ | $0.79\pm0.21$ |
>
>    Here again, explicitly including $\bar{\mathbf{a}}\_{t,\tau}$ in the variational distribution shows a slight performance improvement, though omitting the intervention sequence still maintains performance very close to the optimal value.
>
> Therefore, the approximation bias from omitting $do(\bar{\mathbf{a}}\_{t,\tau})$ has minimal negative impact, primarily because $\mathbf{Y}\_{t+\tau}$ carries effective intervention result information, allowing variational inference to partially capture intervention effects on latent variable $\bar{\mathbf{Z}}\_{t,\tau+1}$ while learning $\mathbf{Y}\_{t+\tau}$.
>
> Additionally, the positivity of $\epsilon\_1$ and $\epsilon\_2$ can be derived from the third inequality in Eq. 22 and Eq. 23, or more rigorously, these values are non-negative, which we will correct in the revised version.

---

### Official Review · Reviewer_whTh · 2025-03-14

**Overall Recommendation:** 3

**Summary:**

This paper presents a new method for finding desirable intervention sequences for individual instances.
First, the authors formulate the task of finding effective intervention sequences as an optimization problem that maximizes the likelihood of the target outcome after the intervention.
Then, the authors propose a framework called VCIP, which uses variational inference to construct a surrogate function that approximates the likelihood of the target outcome.
The numerical experiments demonstrate that the proposed method could effectively find effective intervention sequences to achieve the desired outcomes.

## update after rebuttal

Thank you for your response. I appreciate the authors' efforts to clarify the points I raised. Since my main concerns have been addressed, I maintain my evaluation.

**Claims And Evidence:**

Overall, the claims made in this paper are well-supported by clear and convincing evidence.

**Essential References Not Discussed:**

None in particular.

**Experimental Designs Or Analyses:**

The experiments were generally conducted in a reasonable manner.

However, I have a concern about the Ranking-based Evaluation. In this evaluation, random perturbations are added to the ground truth intervention sequences to generate $k$ new intervention sequences, but the details of the generation process are not clearly described.
As a result, it is unclear to what extent the reported results hold for different levels of perturbation, making the generalizability of this evaluation unclear.

Additionally, as mentioned in ''Methods and Evaluation Criteria'', there is a concern about whether Ranking-based Evaluation is appropriate for discussing the effectiveness of the proposed method.

**Methods And Evaluation Criteria:**

For the proposed framework, the authors provide a clear motivation and rationale for the design of the VCIP framework.

For the numerical experiments, perhaps it may be just my misunderstanding, but I have a slight concern about the ranking-based evaluation in Section 5.1.
In Section 5.1, the authors set $Y_{\\text{target}} = Y[\\bar{a}\_{t,\\tau}]$, and compared whether the model-based ranking of the intervention sequences is consistent with the ground-truth ranking.
I think that \\(\bar{a}\_{t,\tau}\\) is not necessarily an optimal intervention sequence to obtain \\(Y_{\text{target}}\\), so I am not sure what the authors aim to validate through this comparison.

**Other Comments Or Suggestions:**

None in particular.

**Other Strengths And Weaknesses:**

None in particular.

**Questions For Authors:**

1. As mentioned in "Methods and Evaluation Criteria", could you clarify what you aim to demonstrate in Section 5.1? Additionally, why is it reasonable to consider a model desirable if GRP and RCS increase for intervention sequences that are not necessarily optimal or close to optimal for the target outcome?

2. Why are the results of RCS not reported for the MIMIC-III dataset? Was the same evaluation performed for this dataset? If so, could you explain the results?

**Relation To Broader Scientific Literature:**

While existing studies have focused on deriving intervention sequences at the population level, this study proposes a method for optimizing intervention sequences at the individual level. The proposed approach enables learning models that effectively guide decision-making to achieve desirable outcomes. Providing such guidance is increasingly important, particularly in the context of Explainable AI, and is expected to have significant value in fields such as causal inference and algorithmic recourse.

**Theoretical Claims:**

The claim of Theorem 4.1 is somewhat ambiguous.
It might be clearer to modify the claim of Theorem 4.1 to state that the intervention sequence that minimizes Equation (6) is an $\epsilon_1+\epsilon_2$-optimal solution to the problem that maximizes $\mathcal O$.
I  have checked the proof of Theorem 4.1.

---

> ### Author Rebuttal · Authors · 2025-03-31
>
> Thank you for your thorough review and appreciation of our work. Below, we address your main concerns:
>
> **Explanation of GRP and RCS Metrics:**
>
> GRP focuses on how the model ranks a sequence $\bar{a}\_{t,\tau}$ that can definitely achieve $Y\_{target} = Y[\bar{a}\_{t,\tau}]$. Ideally, GRP should be 1. This metric only requires knowledge of the potential outcome of sequence $\bar{a}\_{t,\tau}$, which is available in observational data, making it applicable to real-world datasets (MIMIC-III dataset).
>
> RCS evaluates the model's performance across the entire candidate set, testing whether it can compare the likelihood of different sequences achieving $Y\_{target}$. We use actual target distances for comparison (line 261). However, this requires calculating the true potential outcome for each candidate sequence for each individual, thus it can only be used with simulated datasets (tumor datasets).
>
> Therefore, an increase in GRP indicates the model can more accurately identify sequences that can achieve the target, while an increase in RCS indicates the model can better rank the entire candidate set by likelihood of achieving the target, with higher values showing better alignment with true outcomes.
>
> **Rationale for Setting $Y\_{target} = Y[\bar{a}\_{t,\tau}]$:**
>
> We set $Y\_{target} = Y[\bar{a}\_{t,\tau}]$ to ensure that we can establish that sequence $\bar{a}\_{t,\tau}$ can achieve outcome $Y\_{target}$. In this scenario, an ideal model should rank $\bar{a}\_{t,\tau}$ first among candidate sequences, giving a GRP of 1. Regarding whether $\bar{a}\_t$ is necessarily an optimal intervention sequence to obtain $Y\_{target}$, indeed there may be multiple intervention sequences that can achieve $Y\_{target}$, but this does not affect the principle that an ideal model should rank $\bar{a}\_{t,\tau}$ first among candidate sequences (possibly tied with others).
>
> **Regarding Concerns about Theoretical Claims:**
>
> Thank you for this valuable feedback on Theorem 4.1. We agree that the current claim could be stated more precisely. We appreciate your suggestion to modify the claim to explicitly state that the intervention sequence that minimizes Equation (6) is an $\epsilon$-optimal solution to the problem that maximizes Y.
>
> We will revise Theorem 4.1 in the updated manuscript to clarify this relationship and remove any ambiguity. The formal statement will be adjusted to more accurately reflect the theoretical guarantee provided by our approach.
>
> **Regarding Concerns about the Random Perturbations:**
>
> We employ a hybrid approach to generate candidate sequences. Our framework creates **random sequences** (50%-80% of candidates) and **perturbed ground truth sequences** (20%-50% of candidates).
>
> The perturbation strategy is treatment-mode specific:
> - **For discrete interventions**: We randomly flip bits in the ground truth sequence with probability 0.2.
> - **For continuous interventions**: We apply context-aware shifts where values are modified based on their magnitude (low values shifted up, high values shifted down, middle values shifted randomly).
>
> **Generalizability Considerations**. Our mixed-generation strategy ensures robust evaluation across different perturbation levels by testing against both arbitrary interventions and "near-miss" candidates that challenge the model's discrimination ability. This design ensures our evaluation is generalizable beyond specific perturbation patterns, as it comprehensively tests the model's ability to rank interventions across the similarity spectrum.
>
> Full implementation details are available in our codebase under `src/utils/helper/generate_perturbed_sequences`. We will add relevant details in the updated manuscript.

---

### Decision · Program_Chairs · 2025-05-01

**Decision:**

Accept (poster)

**Comment:**

Short summary: This paper formalizes and studies the problem of “counterfactual target achievement” – in which goal is to find the optimal intervention sequences for a given target in a single instance, while mitigating errors from unobserved counterfactuals.

All reviews recommended a “weak accept” for the paper. Some of the key merits of the paper as identified by reviews include, importance of the problem, novelty of the formulation and contribution, a generally robust experimental analysis (although some suggestions for improvements are made), The reviews also identify some limitations. These include concerns raised around theoretical claims (theorem 4.1), paper being somewhat hard to read and room for improved exposition (improving captions to Tables, etc.)

Furthermore, the rebuttal also provides extensive arguments and explanations to allay many of the reviewer concerns. Hence my reasoning is to accept the paper (medium priority)